# Generalizing Analytic Shrinkage for Arbitrary Covariance Structures

**Daniel Bartz**
Department of Computer Science
TU Berlin, Berlin, Germany
daniel.bartz@tu-berlin.de

**Klaus-Robert Müller**
TU Berlin, Berlin, Germany
Korea University, Korea, Seoul
klaus-robert.mueller@tu-berlin.de

## Abstract

Analytic shrinkage is a statistical technique that offers a fast alternative to cross-validation for the regularization of covariance matrices and has appealing consistency properties. We show that the proof of consistency requires bounds on the growth rates of eigenvalues and their dispersion, which are often violated in data. We prove consistency under assumptions which do not restrict the covariance structure and therefore better match real world data. In addition, we propose an extension of analytic shrinkage –orthogonal complement shrinkage– which adapts to the covariance structure. Finally we demonstrate the superior performance of our novel approach on data from the domains of finance, spoken letter and optical character recognition, and neuroscience.

## 1 Introduction

The estimation of covariance matrices is the basis of many machine learning algorithms and estimation procedures in statistics. The standard estimator is the sample covariance matrix: its entries are unbiased and consistent [1]. A well-known shortcoming of the sample covariance is the systematic error in the spectrum. In particular for high dimensional data, where dimensionality $p$ and number of observations $n$ are often of the same order, large eigenvalues are over- und small eigenvalues underestimated. A form of regularization which can alleviate this bias is shrinkage [2]: the convex combination of the sample covariance matrix $\mathbf{S}$ and a multiple of the identity $\mathbf{T} = p^{-1}\mathrm{trace}(\mathbf{S})\mathbf{I}$,

$$\mathbf{C}^{sh} = (1 - \lambda)\mathbf{S} + \lambda\mathbf{T}, \tag{1}$$

has potentially lower mean squared error and lower bias in the spectrum [3]. The standard procedure for chosing an optimal regularization for shrinkage is cross-validation [4], which is known to be time consuming. For online settings CV can become unfeasible and a faster model selection method is required. Recently, analytic shrinkage [3] which provides a consistent analytic formula for the above regularization parameter $\lambda$ has become increasingly popular. It minimizes the expected mean squared error of the convex combination with a computational cost of $\mathcal{O}(p^2)$, which is negligible when used for algorithms like Linear Discriminant Analysis (LDA) which are $\mathcal{O}(p^3)$.

The consistency of analytic shrinkage relies on assumptions which are rarely tested in practice [5]. This paper will therefore aim to render the analytic shrinkage framework more practical and usable for real world data. We contribute in three aspects: first, we derive simple tests for the applicability of the analytic shrinkage framework and observe that for many data sets of practical relevance the assumptions which underly consistency are not fullfilled. Second, we design assumptions which better fit the statistical properties observed in real world data which typically has a low dimensional structure. Under these new assumptions, we prove consistency of analytic shrinkage. We show a counter-intuitive result: for typical covariance structures, no shrinkage –and therefore no regularization– takes place in the limit of high dimensionality and number of observations. In practice, this leads to weak shrinkage and degrading performance. Therefore, third, we propose an extension of the shrinkage framework: automatic orthogonal complement shrinkage (*aoc-shrinkage*)

takes the covariance structure into account and outperforms standard shrinkage on real world data at a moderate increase in computation time. Note that proofs of all theorems in this paper can be found in the supplemental material.

## 2 Overview of analytic shrinkage

To derive analytic shrinkage, the expected mean squared error of the shrinkage covariance matrix eq. (1) as an estimator of the true covariance matrix $\mathbf{C}$ is minimized:

$$\lambda^\star = \arg\min_\lambda R(\lambda) := \arg\min_\lambda \mathbb{E}\left[\left\|\mathbf{C} - (1-\lambda)\mathbf{S} - \lambda\mathbf{T}\right\|^2\right] \tag{2}$$

$$= \arg\min_\lambda \sum_{i,j}\left\{2\lambda\left\{\mathrm{Cov}\left(S_{ij}, T_{ij}\right) - \mathrm{Var}\left(S_{ij}\right)\right\} + \lambda^2\mathbb{E}\left[\left(S_{ij} - T_{ij}\right)^2\right] + \mathrm{Var}\left(S_{ij}\right)\right\} \tag{3}$$

$$= \frac{\sum_{i,j}\left\{\mathrm{Var}(S_{ij}) - \mathrm{Cov}(S_{ij}, T_{ij})\right\}}{\sum_{i,j}\mathbb{E}\left[\left(S_{ij} - T_{ij}\right)^2\right]}.$$

The analytic shrinkage estimator $\hat{\lambda}$ is obtained by replacing expectations with sample estimates:

$$\widehat{\mathrm{Var}}(S_{ij}) = \frac{1}{(n-1)n}\sum_s\left(x_{is}x_{js} - \frac{1}{n}\sum_t x_{it}x_{jt}\right)^2$$

$$\widehat{\mathrm{Cov}}(S_{ii}, T_{ii}) = \frac{1}{(n-1)np}\sum_k\left\{\sum_s x_{is}^2 x_{ks}^2 - \frac{1}{n}\sum_t x_{it}^2 \sum_{t'} x_{it'}^2\right\}$$

$$\widehat{\mathbb{E}}\left[(S_{ij} - T_{ij})^2\right] = (S_{ij} - T_{ij})^2$$

Theoretical results on the estimator $\hat{\lambda}$ are based on analysis of a sequence of statistical models indexed by $n$. $\mathbf{X}_n$ denotes a $p_n \times n$ matrix of $n$ iid observations of $p_n$ variables with mean zero and covariance matrix $\mathbf{\Sigma}_n$. $\mathbf{Y}_n = \mathbf{\Gamma}_n^{\mathrm{T}}\mathbf{X}_n$ denotes the same observations in their eigenbasis, having diagonal covariance $\mathbf{\Lambda}_n = \mathbf{\Gamma}_n^{\mathrm{T}}\mathbf{\Sigma}_n\mathbf{\Gamma}_n$. Lower case letters $x_{it}^n$ and $y_{it}^n$ denote the entries of $\mathbf{X}^n$ and $\mathbf{Y}^n$, respectively[1]. The main theoretical result on the estimator $\hat{\lambda}$ is its consistency in the large $n, p$ limit [3]. A decisive role is played by an assumption on the eighth moments[2] in the eigenbasis:

**Assumption 2** (**A2**, Ledoit/Wolf 2004 [3]). *There exists a constant $K_2$ independent of $n$ such that*

$$p_n^{-1}\sum_{i=1}^{p_n}\mathbb{E}[(y_{i1}^n)^8] \le K_2.$$

## 3 Implicit assumptions on the covariance structure

From the assumption on the eighth moments in the eigenbasis, we derive requirements on the eigenvalues which facilitate an empirical check:

**Theorem 1** (largest eigenvalue growth rate). *Let **A2** hold. Then, there exists a limit on the growth rate of the largest eigenvalue*

$$\gamma_1^n = \max_i \mathrm{Var}(y_i^n) = O\left(p_n^{1/4}\right).$$

**Theorem 2** (dispersion growth rate). *Let **A2** hold. Then, there exists a limit on the growth rate of the normalized eigenvalue dispersion*

$$d_n = p_n^{-1}\sum_i(\gamma_i - p_n^{-1}\sum_j\gamma_j)^2 = O\left(1\right).$$

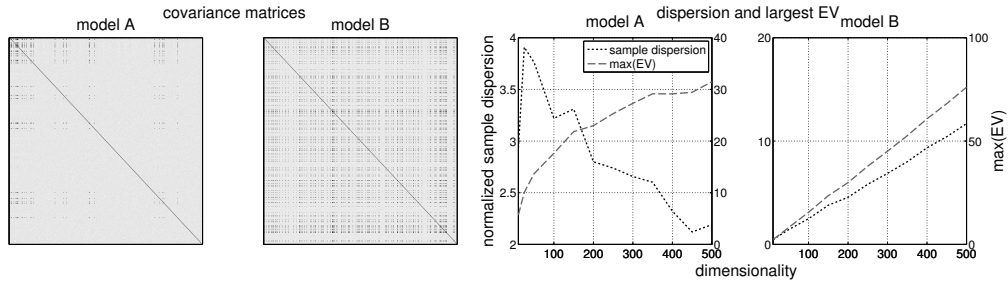

Figure 1: Covariance matrices and dependency of the largest eigenvalue/dispersion on the dimensionality. Average over 100 repetitions.

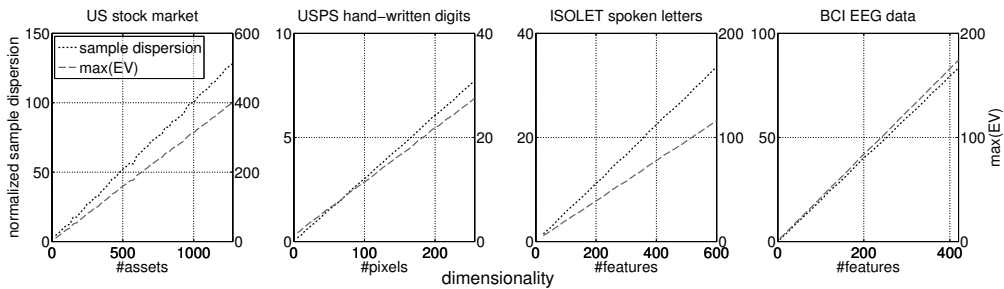

Figure 2: Dependency of the largest eigenvalue/dispersion on the dimensionality. Average over 100 random subsets.

The theorems restrict the covariance structure of the sequence of models when the dimensionality increases. To illustrate this, we design two sequences of models A and B indexed by their dimensionality $p$, in which dimensions $x_i^p$ are correlated with a signal $s^p$:

$$
x_i^p = \begin{cases} (0.5 + b_i^p) \cdot \varepsilon_i^p + \alpha c_i^p s^p, & \text{with probability } P_{s_{A/B}}(i), \\ (0.5 + b_i^p) \cdot \varepsilon_i^p, & \text{else.} \end{cases}
\tag{4}
$$

where $b_i^p$ and $c_i^p$ are uniform random from $[0, 1]$, $s^p$ and $\epsilon_i^p$ are standard normal, $\alpha = 1$, $P_{s_B}(i) = 0.2$ and $P_{s_A}(i) = (i/10 + 1)^{-7/8}$ (power law decay). To avoid systematic errors, we hold the ratio of observations to dimensions fixed: $n^p/p = 2$.

To the left in Figure 1, covariance matrices are shown: For model A, the matrix is dense in the upper left corner, the more dimensions we add the more sparse the matrix gets. For model B, correlations are spread out evenly. To the right, normalized sample dispersion and largest eigenvalue are shown. For model A, we see the behaviour from the theorems: the dispersion is bounded, the largest eigenvalue grows with the fourth root. For model B, there is a linear dependency of both dispersion and largest eigenvalue: **A2** is violated.

For real world data, we measure the dependency of the largest eigenvalue/dispersion on the dimensionality by averaging over random subsets. Figure 2 shows the results for four data sets[3]: (1) New York Stock Exchange, (2) USPS hand-written digits, (3) ISOLET spoken letters and (4) a Brain Computer Interface EEG data set. The largest eigenvalues and the normalized dispersions (see Figure 2) closely resemble model B; a linear dependence on the dimensionality which violates **A2** is visible.

# 4 Analytic shrinkage for arbitrary covariance structures

We replace **A2** by a weaker assumption on the moments in the basis of the observations **X** which does not impose any constraints on the covariance structure[4]:

**Assumption 2′ (A2′).** *There exists a constant $K_2$ independent of p such that*

$$p^{-1} \sum_{i=1}^{p} \mathbb{E}[(x_{i1}^p)^8] \leq K_2.$$

**Standard assumptions**    For the proof of consistency, the relationship between dimensionality and number of observations has to be defined and a weak restriction on the correlation of the products of uncorrelated variables is necessary. We use slightly modified versions of the original assumptions [3].

**Assumption 1′ (A1′, Kolmogorov asymptotics).** *There exists a constant $K_1$, $0 \leq K_1 \leq \infty$ independent of p such that*

$$\lim_{p \to \infty} p/n_p = K_1.$$

**Assumption 3′ (A3′).**

$$\lim_{p \to \infty} \frac{\sum_{i,j,kl,l \in Q_p} \left( \mathrm{Cov}[y_{i1}^p y_{j1}^p, y_{k1}^p y_{l1}^p] \right)^2}{|Q_p|} = 0$$

*where $Q_p$ is the set of all quadruples consisting of distinct integers between 1 and p.*

**Additional Assumptions**    **A1′** to **A3′** subsume a wide range of dispersion and eigenvalue configurations. To investigate the role which this plays, we categorize sequences by adding an additional parameter $k$. It will prove essential for the limit behavior of optimal shrinkage and the consistency of analytic shrinkage:

**Assumption 4 (A4**, growth rate of the normalized dispersion**).** *Let $\gamma_i$ denote the eigenvalues of* **C**. *Then, the limit behaviour of the normalized dispersion is parameterized by $k$:*

$$p^{-1} \sum_{i} (\gamma_i - p^{-1} \sum_{j} \gamma_j)^2 = \Theta \left( \max(1, p^{2k-1}) \right),$$

*where $\Theta$ is the Landau Theta.*

In sequences of models with $k \leq 0.5$ the normalized dispersion is bounded from above and below, as in model A in the last section. For $k > 0.5$ the normalized dispersion grows with the dimensionality, for $k = 1$ it is linear in $p$, as in model B.

We make two technical assumptions to rule out degenerate cases. First, we assume that, on average, additional dimensions make a positive contribution to the mean variance:

**Assumption 5 (A5).** *There exists a constant $K_3$ such that*

$$p^{-1} \sum_{i=1}^{p} \mathbb{E}[(x_{i1}^p)^2] \geq K_3.$$

Second, we assume that limits on the relation between second, fourth and eighth moments exist:

**Assumption 6 (A6**, moment relation**).** $\exists \alpha_4, \alpha_8, \beta_4$ *and* $\beta_8$:

$$
\begin{aligned}
\mathbb{E}[y_i^8] &\leq (1+\alpha_8)\mathbb{E}^2[y_i^4] & \mathbb{E}[y_i^4] &\leq (1+\alpha_4)\mathbb{E}^2[y_i^2] \\
\mathbb{E}[y_i^8] &\geq (1+\beta_8)\mathbb{E}^2[y_i^4] & \mathbb{E}[y_i^4] &\geq (1+\beta_4)\mathbb{E}^2[y_i^2]
\end{aligned}
$$

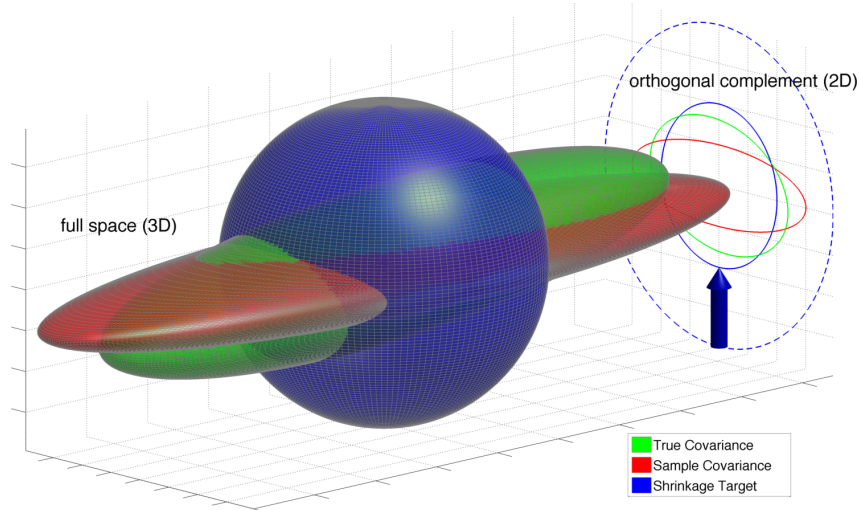

Figure 3: Illustration of orthogonal complement shrinkage.

**Theoretical results on limit behaviour and consistency**   We are able to derive a novel theorem which shows that under these wider assumptions, shrinkage remains consistent:

**Theorem 3** (Consistency of Shrinkage). *Let* **A1′**, **A2′**, **A3′**, **A4**, **A5**, **A6** *hold and*

$$m = \mathbb{E}\left[\left((\lambda^* - \hat{\lambda})/\lambda^*\right)^2\right]$$

*denote the expected squared relative error of the estimate $\hat{\lambda}$. Then, independently of k,*

$$\lim_{p \to \infty} m = 0.$$

An unexpected caveat accompanying this result is the limit behaviour of the optimal shrinkage strength $\lambda^*$:

**Theorem 4** (Limit behaviour). *Let* **A1′**, **A2′**, **A3′**, **A4**, **A5**, **A6** *hold. Then, there exist $0 < b_l < b_u < 1$*

$$
\begin{aligned}
k \le 0.5 &\quad \Rightarrow \quad \forall n : b_l \le \lambda^* \le b_u \\
k > 0.5 &\quad \Rightarrow \quad \lim_{p \to \infty} \lambda^* = 0
\end{aligned}
$$

The theorem shows that there is a fundamental problem with analytic shrinkage: if $k$ is larger than 0.5 (all data sets in the last section had $k = 1$) there is no shrinkage in the limit.

## 5   Automatic orthogonal complement shrinkage

**Orthogonal complement shrinkage**   To obtain a finite shrinkage strength, we propose an extension of shrinkage we call *oc-shrinkage*: it leaves the first eigendirection untouched and performs shrinkage on the orthogonal complement *oc* of that direction. Figure 3 illustrates this approach. It shows a three dimensional true covariance matrix with a high dispersion that makes it highly ellipsoidal. The result is a high level of discrepancy between the spherical shrinkage target and the true covariance. The best convex combination of target and sample covariance will put extremely low weight on the target. The situation is different in the orthogonal complement of the first eigendirection of the sample covariance matrix: there, the discrepancy between sample covariance and target is strongly reduced.

To simplify the theoretical analysis, let us consider the case where there is only a single growing eigenvalue while the remainder stays bounded:

**Assumption 4′** (**A4′** single large eigenvalue)**.** *Let us define*

$$z_i = y_i, \qquad 2 \le i \le p,$$
$$z_1 = p^{-k/2}y_1.$$

*There exist constants $F_l$ and $F_u$ such that $F_l \le \mathbb{E}[z_i^8] \le F_u$*

A recent result from Random Matrix Theory [6] allows us to prove that the projection on the *empirical* orthogonal complement $\widehat{oc}$ does not affect the consistency of the estimator $\hat{\lambda}_{\widehat{oc}}$:

**Theorem 5** (consistency of oc-shrinkage)**.** *Let* **A1′**, **A2′**, **A3′**, **A4′**, **A5**, **A6** *hold. In addition, assume that $16^{th}$ moments[5] of the $y_i$ exist and are bounded. Then, independently of $k$,*

$$\lim_{p \to \infty} \left( \hat{\lambda}_{\widehat{oc}} - \arg\min_{\lambda} Q_{\widehat{oc}}(\lambda) \right)^2 = 0,$$

*where $Q$ denotes the mean squared error (MSE) of the convex combination (cmp. eq. (2)).*

**Automatic model selection**   Orthogonal complement shrinkage only yields an advantage if the first eigenvalue is large enough. Starting from eq. (2), we can consistently estimate the error of standard shrinkage and orthogonal complement shrinkage and only use oc-shrinkage when the difference $\widehat{\Delta}_{R,\widehat{oc}}$ is positive. In the supplemental material, we derive a formula of a conservative estimate:

$$\widehat{\Delta}_{R,cons.,\widehat{oc}} = \widehat{\Delta}_{R,\widehat{oc}} - m_\Delta \hat{\sigma}_{\widehat{\Delta}_{R,\widehat{oc}}} - m_E \hat{\lambda}_{\widehat{oc}}^2 \hat{\sigma}_{\hat{E}}.$$

Usage of $m_\Delta = 0.45$ corresponds to 75% probability of improvement under gaussianity and yields good results in practice. The second term is relevant in small samples, setting $m_E = 0.1$ is sufficient. A dataset may have multiple large eigenvalues. It is straightforward to iterate the procedure and thus automatically select the number of retained eigendirections $\hat{r}$. We call this *automatic orthogonal complement shrinkage*. An algorithm listing can be found in the supplemental.

The computational cost of aoc-shrinkage is larger than that of standard shrinkage as it additionally requires an eigendecomposition $\mathcal{O}(p^3)$ and some matrix multiplications $\mathcal{O}(\hat{r}p^2)$. In the applications considered here, this additional cost is negligible: $\hat{r} \ll p$ and the eigendecomposition can replace matrix inversions for LDA, QDA or portfolio optimization.

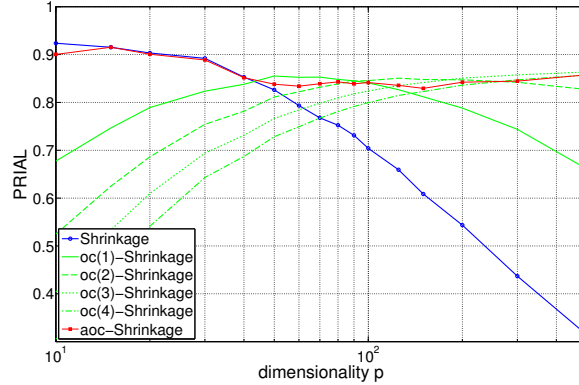

Figure 4: Automatic selection of the number of eigendirections. Average over 100 runs.

## 6  Empirical validation

**Simulations**   To test the method, we extend model B (eq. (4), section 3) to three signals, $P_{s_i} = (0.1, 0.25, 0.5)$. Figure 4 reports the percentage improvement in average loss over the sample covariance matrix,

$$\mathrm{PRIAL}\big(\mathbf{C}^{sh/oc-sh/aoc-sh}\big) = \frac{\mathbb{E}\|\mathbf{S} - \mathbf{C}\| - \mathbb{E}\|\mathbf{C}^{sh/oc-sh/aoc-sh} - \mathbf{C}\|}{\mathbb{E}\|\mathbf{S} - \mathbf{C}\|},$$

Table 1: Portfolio risk. Mean absolute deviations$\cdot 10^3$ (mean squared deviations$\cdot 10^6$) of the resulting portfolios for the different covariance estimators and markets. $\dagger := $ *aoc-shrinkage* significantly better than this model at the 5% level, tested by a randomization test.

|  | US | EU | HK |
|---|---|---|---|
| sample covariance | $8.56^\dagger$ ($156.1^\dagger$) | $5.93^\dagger$ ($78.9^\dagger$) | $6.57^\dagger$ ($81.2^\dagger$) |
| standard shrinkage | $6.27^\dagger$ ($86.4^\dagger$) | $4.43^\dagger$ ($46.2^\dagger$) | $6.32^\dagger$ ($76.2^\dagger$) |
| $\hat{\lambda}$ | 0.09 | 0.12 | 0.10 |
| shrinkage to a factor model | $5.56^\dagger$ ($69.6^\dagger$) | $4.00^\dagger$ ($39.1^\dagger$) | $6.17^\dagger$ ($72.9^\dagger$) |
| $\hat{\lambda}$ | 0.41 | 0.44 | 0.42 |
| aoc-shrinkage | **5.41** (**67.0**) | **3.83** (**36.3**) | **6.11** (**71.8**) |
| $\hat{\lambda}$ | 0.75 | 0.79 | 0.75 |
| average $\hat{r}$ | 1.64 | 1.17 | 1.41 |

Table 2: Accuracies for classification tasks on ISOLET and USPS data. $* := $ significantly better than all compared methods at the 5% level, tested by a randomization test.

| | ISOLET | | | USPS | | |
|---|---|---|---|---|---|---|
| $n_{train}$ | 500 | 2000 | 5000 | 500 | 2000 | 5000 |
| LDA | 75.77% | 92.29% | 94.1% | 72.31% | 87.45% | 89.56% |
| LDA (shrinkage) | 88.92% | 93.25% | 94.3% | 83.77% | 88.37% | 89.77% |
| LDA (aoc) | **89.69**%$^*$ | **93.42**%$^*$ | **94.33**%$^*$ | **83.95**%$^*$ | 88.37% | 89.77% |
| QDA | 2.783% | 4.882% | 14.09% | 10.11% | 49.45% | 72.43% |
| QDA (shrinkage) | 58.57% | 75.4% | 79.25% | 82.2% | 88.85% | 89.67% |
| QDA (aoc) | 59.51% | 80.84% | 87.35% | 83.31% | **89.4**%$^*$ | **90.07**% |

of standard shrinkage, oc-shrinkage for one to four eigendirections and aoc-shrinkage.

Standard shrinkage behaves as predicted by Theorem 4: $\hat{\lambda}$ and therefore the PRIAL tend to zero in the large $n$, $p$ limit. The same holds for orders of *oc-shrinkage* –oc(1) and oc(2)– lower than the number of signals, but performance degrades more slowly. For small dimensionalities eigenvalues are small and therefore there is no advantage for oc-shrinkage. On the contrary, the higher the order of oc-shrinkage, the larger the error by projecting out spurious large eigenvalues which should have been subject to regularization. The automatic order selection *aoc-shrinkage* leads to close to optimal PRIAL for all dimensionalities.

**Real world data I: portfolio optimization**   Covariance estimates are needed for the minimization of portfolio risk [7]. Table 1 shows portfolio risk for approximately eight years of daily return data from 1200 US, 600 European and 100 Hong Kong stocks, aggregated from Reuters tick data [8]. Estimation of covariance matrices is based on short time windows (150 days) because of the data's nonstationarity. Despite the unfavorable ratio of observations to dimensionality, standard shrinkage has very low values of $\hat{\lambda}$: the stocks are highly correlated and the spherical target is highly inappropriate. Shrinkage to a financial factor model incorporating the market factor [9] provides a better target; it leads to stronger shrinkage and better portfolios. Our proposed *aoc-shrinkage* yields even stronger shrinkage and significantly outperforms all compared methods.

Table 3: Accuracies for classification tasks on BCI data. Artificially injected noise in one electrode. $* := $ significantly better than all compared methods at the 5% level, tested by a randomization test.

| $\sigma_{noise}$ | 0 | 10 | 30 | 100 | 300 | 1000 |
|---|---|---|---|---|---|---|
| LDA | 92.28% | 92.28% | 92.28% | 92.28% | 92.28% | 92.28% |
| LDA (shrinkage) | 92.39% | 92.94% | 92.18% | 88.04% | 82.15% | 73.79% |
| LDA (aoc) | **93.27**%$^*$ | **93.27**%$^*$ | **93.24**%$^*$ | **92.88**%$^*$ | **93.16**%$^*$ | **93.19**%$^*$ |
| average $\hat{r}$ | 2.0836 | 3.0945 | 3.0891 | 3.0891 | 3.0891 | 3.09 |

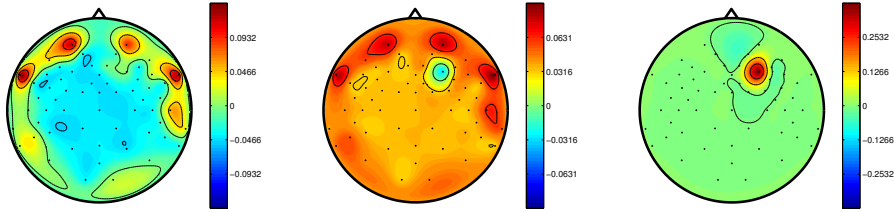

Figure 5: High variance components responsible for failure of shrinkage in BCI. $\sigma_{noise} = 10$. Subject 1.

**Real world data II: USPS and ISOLET**    We applied Linear and Quadratic Discriminant Analysis (LDA and QDA) to hand-written digit recognition (USPS, 1100 observations with 256 pixels for each of the 10 digits [10]) and spoken letter recognition (ISOLET, 617 features, 7797 recordings of 26 spoken letters [11], obtained from the UCI ML Repository [12]) to assess the quality of standard and *aoc*-shrinkage covariance estimates.

Table 2 shows that aoc-shrinkage outperforms standard shrinkage for QDA and LDA on both data sets for different training set sizes. Only for LDA and large sample sizes on the relatively low dimensional USPS data, there is no difference between standard and aoc-shrinkage: the automatic procedure decides that shrinkage on the whole space is optimal.

**Real world data III: Brain-Computer-Interface**    The BCI data was recorded in a study in which 11 subjects had to distinguish between noisy and noise-free phonemes [13, 14]. We applied LDA on 427 standardized features calculated from event related potentials in 61 electrodes to classify two conditions: correctly identified noise-free and correctly identified noisy phonemes ($n_{train} = 1000$).

For Table 3, we simulated additive noise in a random electrode (100 repetitions). With and without noise, our proposed aoc-shrinkage outperforms standard shrinkage LDA. Without noise, $\hat{r} \approx 2$ high variance directions –probably corresponding to ocular and facial muscle artefacts, depicted to the left in Figure 5– are left untouched by aoc-shrinkage. With injected noise, the number of directions increases to $\hat{r} \approx 3$, as the procedure detects the additional high variance component –to the right in Figure 5– and adapts the shrinkage procedure such that performance remains unaffected. For standard shrinkage, noise affects the analytic regularization and performance degrades as a result.

## 7    Discussion

Analytic shrinkage is a fast and accurate alternative to cross-validation which yields comparable performance, e.g. in prediction tasks and portfolio optimization. This paper has contributed by clarifying the (limited) applicability of the analytic shrinkage formula. In particular we could show that its assumptions are often violated in practice since real world data has complex structured dependencies. We therefore introduced a set of more general assumptions to shrinkage theory, chosen such that the appealing consistency properties of analytic shrinkage are preserved. We have shown that for typcial structure in real world data, strong eigendirections adversely affect shrinkage by driving the shrinkage strength to zero. Therefore, finally, we have proposed an algorithm which automatically restricts shrinkage to the orthogonal complement of the strongest eigendirections if appropriate. This leads to improved robustness and significant performance enhancement in simulations and on real world data from the domains of finance, spoken letter and optical character recognition, and neuroscience.

**Acknowledgments**

This work was supported in part by the World Class University Program through the National Research Foundation of Korea funded by the Ministry of Education, Science, and Technology, under Grant R31-10008. We thank Gilles Blanchard, Duncan Blythe, Thorsten Dickhaus, Irene Winkler and Anne Porbadnik for valuable comments and discussions.

## Footnotes

[1]We shall often drop the sequence index $n$ and the observation index $t$ to improve readability of formulas.

[2]eighth moments arise because $\mathrm{Var}(S_{ij})$, the variance of the sample covariance, is of fourth order and has to converge. Nevertheless, even for for non-Gaussian data convergence is fast.

[3]for details on the data sets, see section 5.

[4]For convenience, we index the sequence of statistical models by $p$ instead of $n$.

[5]The existence of $16^{th}$ moments is needed because we bound the estimation error in each direction by the maximum over all directions, an extremely conservative approximation.

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
