[Supplementary Material]

# Supplemental Material

# Generalizing Analytic Shrinkage for Arbitrary Covariance Structures

# Proofs, Algorithm and Derivations

**Daniel Bartz**
Department of Computer Science
TU Berlin, Berlin, Germany
`daniel.bartz@tu-berlin.de`

**Klaus-Robert Müller**
TU Berlin, Berlin, Germany
Korea University, Korea, Seoul
`klaus-robert.mueller@tu-berlin.de`

## Contents

# 1 Proofs

## 1.1 Proof of Theorem 1

**Theorem 1** (largest eigenvalue growth rate)**.**

*Proof.* Under assumption **A2**, we have

$$K_2 \geq \frac{1}{p_n} \sum_{i=1}^{p_n} \mathbb{E}[(y_{i1}^n)^8] \geq \frac{1}{p_n} \sum_{i=1}^{p_n} \mathbb{E}^4[(y_{i1}^n)^2] \geq \frac{\gamma_1^4}{p_n}$$

$$\iff \quad \gamma_1 \leq p_n^{\frac{1}{4}} K_2^{\frac{1}{4}}.$$

Let us assume (1) $p_n = n$ and (2) $y_1^n$ is a zero mean gaussian variable with $\mathbb{E}[(y_1^n)^2] = \sqrt[4]{p_n}$, while $y_{i>1}^n = 0$. In this model, **A2** holds:

$$\frac{1}{p_n} \sum_i \mathbb{E}[(y_i^n)^8] = 105.$$

For the largest eigenvalue we have $\gamma_1^n = \sqrt[4]{p_n}$, which attains the above rate and thereby proves that no lower rate exists for all sequences. □

## 1.2 Proof of Theorem 2

**Theorem 2** (dispersion growth rate)**.**

*Proof.* Under assumption **A2**, we have

$$d_n = \frac{1}{p_n} \sum_i (\gamma_i - \sum_j \gamma_j)^2 \leq \frac{1}{p_n} \sum_i \gamma_i^2 \leq \sqrt{\frac{1}{p_n} \sum_i \gamma_i^4} \leq \sqrt{\frac{1}{p_n} \sum_i \mathbb{E}[(y_i^n)^8]} \leq \sqrt{K_2}.$$

Now assume we have gaussian data with

$$\gamma_i = \begin{cases} c & \text{for } i \leq p_n/2 \\ 0 & \text{for } i > p_n/2. \end{cases}$$

There is a $c$ for which **A2** is fulfilled:

$$\frac{1}{p_n} \sum_i \mathbb{E}[(y_i^n)^8] = \frac{1}{p_n} 105 \sum_i \gamma_i = \frac{105}{2} c \leq K_2.$$

For the dispersion, we have

$$d_n = \frac{1}{p_n} \sum_i (\gamma_i - \sum_j \gamma_j)^2 = \frac{1}{p_n} \frac{p_n}{2} (c - \frac{1}{2}c)^2 = \frac{1}{8} c^2.$$

The above rate is attained and therefore no lower rate exists which bounds all sequences.

□

## 1.3 Proof of Theorem 3

Note that Theorem 3 makes use of Theorem 4. Let us first state some helpful properties of the estimators appearing in $\hat{\lambda}$:

**Lemma 1.** $\widehat{\mathrm{Var}}(S_{ij})$, $\widehat{\mathrm{Var}}(T_{ij})$, $\sum_{ij} \widehat{\mathrm{Cov}}(S_{ij}, T_{ij})$ and $(S_{ij} - T_{ij})^2$ *are unbiased estimators of* $\mathrm{Var}(S_{ij})$, $\mathrm{Var}(T_{ij})$, $\sum_{ij} \mathrm{Cov}(S_{ij}, T_{ij})$ *and* $\mathbb{E}[(S_{ij} - T_{ij})^2]$.

*Proof.* follows directly by calculation of the expectations. □

**Theorem 3** (Consistency of Shrinkage)**.**

The proof of consistency under the new assumptions is similar to the standard shrinkage proof in [1]. An additional ingredient is the k-dependent limit behaviour from Theorem 4, which allows us to find bounds on specific terms which would otherwise not converge to zero in the limit.

*Proof.* in a first step, let us rewrite the relative error:

$$m = \left( \frac{\lambda^* - \hat{\lambda}}{\lambda^*} \right)^2 = \left( 1 - \frac{\hat{\lambda}}{\lambda^*} \right)^2.$$

For $p \to \infty$, we have

$$\lim_{p \to \infty} m = \lim_{p \to \infty} \left( 1 - \frac{\hat{\lambda}}{\lambda^*} \right)^2$$

$$= \lim_{p \to \infty} \left( 1 - \left( \frac{n_p p^{-2} \sum_{ij} \widehat{\mathrm{Var}}(S'_{ij})}{p^{-\max(1,2k)} \sum_{ij} (S'_{ij} - T_{ij})^2} \right) \cdot \left( \frac{n_p p^{-2} \sum_{ij} \mathrm{Var}(S'_{ij})}{p^{-\max(1,2k)} \sum_{ij} \mathbb{E}[(S'_{ij} - T_{ij})^2]} \right)^{-1} \right)^2.$$

Here, the factors which have been introduced make each expression bounded from above and below. Let us now look at the convergences of these normalized estimators.

**Bound on the Variance of the Dispersion**    The dispersion is

$$\frac{1}{p^{\max(1,2k)}} \sum_{ij} (S_{ij} - T_{ij})^2 = \frac{1}{p^{\max(1,2k)}} \sum_{ij} S_{ij}^2 - 2 S_{ij} T_{ij} + T_{ij}^2.$$

Let us here only analyse the first term on the r.h.s. which has the highest variance, for the other terms the proof is similar but easier:

$$\frac{1}{p^{\max(1,2k)}} \sum_{ij} S_{ij}^2 = \frac{1}{p^{\max(1,2k)}} \sum_{ij} \left( \frac{1}{n} \sum_s y_{is} y_{jt} \right)^2 = \frac{p^2}{p^{\max(1,2k)} n^2} \sum_{st} \left( \frac{1}{p} \sum_i y_{is} y_{it} \right)^2$$

$$= \frac{p^2}{p^{\max(1,2k)} n^2} \sum_s \left( \frac{1}{p} \sum_i y_{is}^2 \right)^2 + \frac{p^2}{p^{\max(1,2k)} n^2} \sum_{s,t \neq s} \left( \frac{1}{p} \sum_i y_{is} y_{it} \right)^2. \tag{1}$$

We show that the variance goes to zero by showing that both terms on the right-hand side have zero variance in the limit. For the first term, we have:

$$\mathrm{Var}\left( \frac{p^2}{p^{\max(1,2k)} n^2} \sum_s \left( \frac{1}{p} \sum_i y_{is}^2 \right)^2 \right) \leq \frac{p^4}{p^{\max(2,4k)} n^3} \mathbb{E}\left[ \left( \frac{1}{p} \sum_i y_{i1}^2 \right)^4 \right]$$

$$= \frac{p^4}{p^{\max(2,4k)} n^3} \mathbb{E}\left[ \left( \frac{1}{p} \sum_i x_{i1}^2 \right)^4 \right] \leq \frac{p^4}{p^{\max(2,4k)} n^3} \mathbb{E}\left[ \frac{1}{p} \sum_i x_{i1}^8 \right]$$

$$\leq \frac{p^4}{p^{\max(2,4k)} n^3} K_2 \to 0.$$

Therefore the first term in eq. (1) converges to it expectation. Let us now look at the second term:

$$\mathrm{Var}\left( \frac{p^2}{p^{\max(1,2k)} n^2} \sum_{s,t \neq s} \left( \frac{1}{p} \sum_i y_{is} y_{it} \right)^2 \right)$$

$$= \frac{p^4}{p^{\max(2,4k)} n^4} \sum_{s,t \neq s} \sum_{s',t' \neq s'} \mathrm{Cov}\left( \left( \frac{1}{p} \sum_i y_{is} y_{it} \right)^2, \left( \frac{1}{p} \sum_i y_{is} y_{it} \right)^2 \right). \tag{2}$$

The covariance expression only depends on the cardinal of the intersection, which we denote by $(\{s,t\} \cup \{s',t'\})^{\#}$ and which can take the values of 0, 1 and 2. When this cardinality is zero,

$$(\{s,t\} \cup \{s',t'\})^{\#} = 0,$$

there is independence and the covariance is zero as well. For

$$(\{s,t\} \cup \{s',t'\})^{\#} = 1,$$

we have $4n(n-1)(n-2)$ epxressions of the form

$$\text{Cov}\left(\left(\frac{1}{p}\sum_i y_{is}y_{it}\right)^2, \left(\frac{1}{p}\sum_i y_{is}y_{it}\right)^2\right) = \text{Cov}\left(\left(\frac{1}{p}\sum_i y_{i1}y_{i2}\right)^2, \left(\frac{1}{p}\sum_i y_{i1}y_{i3}\right)^2\right)$$

$$= \mathbb{E}\left[\left(\frac{1}{p}\sum_i y_{i1}y_{i2}\right)^2 \left(\frac{1}{p}\sum_i y_{i1}y_{i3}\right)^2\right] - \mathbb{E}\left[\left(\frac{1}{p}\sum_i y_{i1}y_{i2}\right)^2\right]\mathbb{E}\left[\left(\frac{1}{p}\sum_i y_{i1}y_{i3}\right)^2\right]$$

$$\leq \max\left(\mathbb{E}\left[\left(\frac{1}{p}\sum_i y_{i1}y_{i2}\right)^2\left(\frac{1}{p}\sum_i y_{i1}y_{i3}\right)^2\right], \mathbb{E}^2\left[\left(\frac{1}{p}\sum_i y_{i1}y_{i2}\right)^2\right]\right),$$

as both terms are positive. For the first term, we have

$$\mathbb{E}\left[\left(\frac{1}{p}\sum_i y_{i1}y_{i2}\right)^2\left(\frac{1}{p}\sum_i y_{i1}y_{i3}\right)^2\right] = \frac{1}{p^4}\sum_{i,j,i',j'}\mathbb{E}\left[y_{i1}y_{i'1}y_{j1}y_{j'1}\right]\mathbb{E}\left[y_{i2}y_{i'2}\right]\mathbb{E}\left[y_{j3}y_{j'3}\right]$$

$$= \frac{1}{p^4}\sum_{i,j}\mathbb{E}\left[y_{i1}^2y_{j1}^2\right]\mathbb{E}\left[y_{i2}^2\right]\mathbb{E}\left[y_{j3}^2\right] \leq \frac{1}{p^2}\left(\frac{1}{p}\sum_i\sqrt{\mathbb{E}[y_{i1}^4]}\mathbb{E}\left[y_{i2}^2\right]\right)^2$$

$$\overset{\text{A6}}{\leq} \frac{1+\alpha_4}{p^2}\left(\frac{1}{p}\sum_i\mathbb{E}^2\left[y_{i2}^2\right]\right)^2 = \frac{1+\alpha_4}{p^2}\Theta\left(\max(1,p^{4k-2})\right).$$

For the second term, we have

$$\mathbb{E}^2\left[\left(\frac{1}{p}\sum_i y_{i1}y_{i2}\right)^2\right] = \left(\frac{1}{p^2}\sum_{i,j}\mathbb{E}^2\left[y_{i1}y_{j1}\right]\right)^2 = \frac{1}{p^2}\left(\frac{1}{p}\sum_i\mathbb{E}^2\left[y_{i1}^2\right]\right)^2 = \frac{1}{p^2}\Theta\left(\max(1,p^{4k-2})\right)$$

Therefore, we have, combined with the prefactors,

$$\frac{p^4 4n(n-1)(n-2)}{p^{\max(2,4k)}n^4}\left|\text{Cov}\left(\left(\frac{1}{p}\sum_i y_{is}y_{it}\right)^2, \left(\frac{1}{p}\sum_i y_{is}y_{it}\right)^2\right)\right|$$

$$= \frac{4p^4}{p^{\max(2,4k)}n}\frac{1}{p^2}\mathcal{O}\left(\max(1,p^{4k-2})\right) = \frac{1}{n}\mathcal{O}(1) \to 0,$$

and we have shown that for the terms with $(\{s,t\} \cup \{s',t'\})^{\#} = 1$, the variance goes to zero.

For

$$(\{s,t\} \cup \{s',t'\})^{\#} = 2,$$

we get $2n(n-1)$ expressions of the form

$$\left|\text{Cov}\left(\left(\frac{1}{p}\sum_i y_{is}y_{it}\right)^2, \left(\frac{1}{p}\sum_i y_{is}y_{it}\right)^2\right)\right| = \left|\text{Cov}\left(\left(\frac{1}{p}\sum_i y_{i1}y_{i2}\right)^2, \left(\frac{1}{p}\sum_i y_{i1}y_{i2}\right)^2\right)\right|$$

$$\leq \frac{1}{p^4}\sum_{i,j,i',j'}\left|\text{Cov}\left(y_{i1}y_{i2}y_{i'1}y_{i'2}, y_{j1}y_{j2}y_{j'1}y_{j'2}\right)\right|$$

In this summation, we decompose the set of integers into two disjoint subsets: $\{1, \ldots, p\}^4 = Q \cup R$, where $Q$ is the set of distinct integers and $R$ is the remainder:

$$= \frac{1}{p^4} \sum_{i,j,i',j' \in Q} |\text{Cov}\left(y_{i1}y_{i2}y_{i'1}y_{i'2}, y_{j1}y_{j2}y_{j'1}y_{j'2}\right)| + \frac{1}{p^4} \sum_{i,j,i',j' \in R} |\text{Cov}\left(y_{i1}y_{i2}y_{i'1}y_{i'2}, y_{j1}y_{j2}y_{j'1}y_{j'2}\right)|.$$

For the sum over $Q$, we obtain can bring this into a form which is zero as a consequence of **A3′**:

$$|\text{Cov}\left(y_{i1}y_{i2}y_{i'1}y_{i'2}, y_{j1}y_{j2}y_{j'1}y_{j'2}\right)|$$
$$= \left|\mathbb{E}^2\left[y_{i1}y_{i'1}y_{j1}y_{j'1}\right] - \mathbb{E}^2\left[y_{i1}y_{i'1}\right]\mathbb{E}^2\left[y_{i2}y_{i'2}\right]\right| = \mathbb{E}^2\left[y_{i1}y_{i'1}y_{j1}y_{j'1}\right]$$
$$= \left(\text{Cov}\left(y_{i1}y_{i'1}, y_{j1}y_{j'1}\right) + \mathbb{E}\left[y_{i1}y_{i'1}\right]\mathbb{E}\left[y_{j1}y_{j'1}\right]\right)^2 = \left(\text{Cov}\left(y_{i1}y_{i'1}, y_{j1}y_{j'1}\right)\right)^2. \qquad (3)$$

Taking the prefactors into account, we get

$$\frac{p^4 2n(n-1)}{p^{\max(2,4k)}n^4} \frac{1}{p^4} \sum_{(i,j,i',j') \in Q} |\text{Cov}\left(y_{i1}y_{i2}y_{i'1}y_{i'2}, y_{j1}y_{j2}y_{j'1}y_{j'2}\right)|$$

$$\leq 48 \frac{p^4}{p^{\max(2,4k)}n^2} \sum_{(i,j,i',j') \in Q} \frac{\left(\text{Cov}\left(y_{i1}y_{i'1}, y_{j1}y_{j'1}\right)\right)^2}{|Q_p|}$$

$$= \mathcal{O}(1) \cdot \sum_{(i,j,i',j') \in Q} \frac{\left(\text{Cov}\left(y_{i1}y_{i'1}, y_{j1}y_{j'1}\right)\right)^2}{|Q_p|} \overset{\text{A3′}}{\to} 0.$$

For the sum over $R$, we have

$$\frac{1}{p^4} \sum_{(i,j,i',j') \in R} |\text{Cov}\left(y_{i1}y_{i2}y_{i'1}y_{i'2}, y_{j1}y_{j2}y_{j'1}y_{j'2}\right)|$$

$$\leq \frac{1}{p^4} \sum_{i,j,j'} 2\left|\text{Cov}\left(y_{i1}^2 y_{i2}^2, y_{j1}y_{j2}y_{j'1}y_{j'2}\right)\right| + 4\left|\text{Cov}\left(y_{i1}y_{i2}y_{i'1}y_{i'2}, y_{i1}y_{i2}y_{j1}y_{j2}\right)\right|$$

$$\leq \frac{1}{p^4} \sum_{i,j,j'} 2\sqrt{\mathbb{E}\left[y_{i1}^4 y_{i2}^4\right]\mathbb{E}\left[y_{j1}^2 y_{j2}^2 y_{j'1}^2 y_{j'2}^2\right]} + 4\sqrt{\mathbb{E}\left[y_{i1}^2 y_{i2}^2 y_{j1}^2 y_{j2}^2\right]\mathbb{E}\left[y_{i1}^2 y_{i2}^2 y_{j'1}^2 y_{j'2}^2\right]}$$

$$\leq \frac{6}{p^4} \sum_{i,j,j'} \mathbb{E}\left[y_{i1}^4\right]\sqrt{\mathbb{E}\left[y_{j1}^4\right]}\sqrt{\mathbb{E}\left[y_{j'1}^4\right]} \overset{\text{A6}}{\leq} \frac{6(1+\alpha_4)}{p}\left(\frac{1}{p}\sum_i \mathbb{E}^2\left[y_{i1}^2\right]\right)\left(\frac{1}{p}\sum_j \mathbb{E}\left[y_{j1}^2\right]\right)^2$$

$$= \frac{6(1+\alpha_4)\sqrt{K_2}}{p}\Theta\left(\max(1, p^{2k-1})\right). \qquad (4)$$

Together with the prefactors, we obtain

$$\frac{p^4 2n(n-1)}{p^{\max(2,4k)}n^4} \frac{1}{p^4} \sum_{(i,j,i',j') \in R} |\text{Cov}\left(y_{i1}y_{i2}y_{i'1}y_{i'2}, y_{j1}y_{j2}y_{j'1}y_{j'2}\right)|$$

$$= \frac{2}{p^{\max(1,4k-1)}n^2}\mathcal{O}\left(\max(1, p^{2k-1})\right) \to 0.$$

and we have shown that for the terms with $(\{s,t\} \cup \{s',t'\})^{\#} = 2$, the variance goes to zero.

**Bound on the variance of** $np^{-2}\sum_{ij} \widehat{\text{Var}}(S_{ij})$    Let us first rewrite the sample estimate:

$$\frac{n}{p^2}\widehat{\text{Var}}\left(S'_{ij}\right) = \frac{n}{p^2}\left(\frac{1}{(n-1)n}\sum_s \left(y_{is}y_{js} - \frac{1}{n}\sum_{s'} y_{is'}y_{js'}\right)^2\right)$$

$$= \frac{1}{(n-1)p^2}\left(\sum_s y_{is}^2 y_{js}^2 - \frac{1}{n}\sum_{s,s'} y_{is}y_{js}y_{is'}y_{js'}\right)$$

$$= \frac{1}{(n-1)p^2} \left( \sum_n y_{in}^2 y_{jn}^2 - nS_{ij}^2 \right).$$

Let us now have a look at the second term. In the last paragraph, we have already shown that the variance of

$$\frac{1}{p^{\max(1,2k)}} \sum_{ij} S_{ij}^2$$

goes to zero, which just differs by a larger prefactor.

For the first term, we have

$$\mathrm{Var}\left( \frac{1}{(n-1)p^2} \sum_{ij} \sum_n y_{in}^2 y_{jn}^2 \right) = \frac{n}{(n-1)^2 p^4} \sum_{i,i',j,j'} \mathrm{Cov}\left( y_{i1}^2 y_{j1}^2, y_{i'1}^2 y_{j'1}^2 \right)$$

$$= \frac{n}{(n-1)^2 p^4} \sum_{(i,i',j,j') \in Q} \mathrm{Cov}\left( y_{i1}^2 y_{j1}^2, y_{i'1}^2 y_{j'1}^2 \right) + \frac{n}{(n-1)^2 p^4} \sum_{(i,i',j,j') \in R} \mathrm{Cov}\left( y_{i1}^2 y_{j1}^2, y_{i'1}^2 y_{j'1}^2 \right).$$

We could do a proof without separating terms, but as we need the results later, we again decompose the set of integers. If all integers are distinct, we obtain

$$\frac{n}{(n-1)^2 p^4} \sum_{(i,i',j,j') \in Q} \mathrm{Cov}\left( y_{i1}^2 y_{j1}^2, y_{i'1}^2 y_{j'1}^2 \right)$$

$$\leq \frac{n}{(n-1)^2 p^4} \sum_{(i,i',j,j') \in Q} \sqrt{\mathbb{E}[y_{i1}^4 y_{j1}^4]} \sqrt{\mathbb{E}[y_{i'1}^4 y_{j'1}^4]}$$

$$\leq \frac{n}{(n-1)^2 p^4} \sum_{(i,i',j,j') \in Q} \sqrt[4]{\mathbb{E}[y_{i1}^8] \mathbb{E}[y_{j1}^8]} \sqrt[4]{\mathbb{E}[y_{i'1}^8] \mathbb{E}[y_{j'1}^8]}$$

$$\leq \frac{n}{(n-1)^2} \left( \frac{1}{p} \sum_i \mathbb{E}[y_{i1}^2] \right)^4 \leq \frac{n}{(n-1)^2} K_2 \to 0.$$

For $(i, i', j, j') \in R$ we have

$$\frac{n}{(n-1)^2 p^4} \sum_{(i,i',j,j') \in R} \mathrm{Cov}\left( y_{i1}^2 y_{j1}^2, y_{i'1}^2 y_{j'1}^2 \right)$$

$$\leq \frac{6n}{(n-1)^2 p^4} \sum_{i,j,i'} \mathrm{Cov}\left( y_{i1}^2 y_{j1}^2, y_{i'1}^2 y_{i1}^2 \right) + \mathrm{Cov}\left( y_{i1}^2 y_{j1}^2, y_{i'1}^4 \right)$$

$$\leq \frac{6n}{(n-1)^2 p^4} \sum_{i,j,i'} \sqrt{\mathbb{E}[y_{i1}^4 y_{j1}^4]} \sqrt{\mathbb{E}[y_{i'1}^4 y_{i1}^4]} + \sqrt{\mathbb{E}[y_{i1}^4 y_{j1}^4]} \sqrt{\mathbb{E}[y_{i'1}^8]}$$

$$\leq \frac{6n}{(n-1)^2 p^4} \sum_{i,j,i'} \sqrt[4]{\mathbb{E}[y_{i1}^8] \mathbb{E}[y_{j1}^8]} \sqrt[4]{\mathbb{E}[y_{i'1}^8] \mathbb{E}[y_{i1}^8]} + \sqrt[4]{\mathbb{E}[y_{i1}^8] \mathbb{E}[y_{j1}^8]} \sqrt{\mathbb{E}[y_{i'1}^8]}$$

$$\leq \frac{12n(1+\alpha_8)}{(n-1)^2 p^4} \sum_{i,j,i'} \mathbb{E}[y_{i1}^2] \mathbb{E}[y_{j1}^2] \mathbb{E}^2[y_{i'1}^2] \leq \frac{6n}{(n-1)^2 p} K_2 \mathcal{O}\left( \max(1, p^{2k-1}) \right) \to 0.$$

$$(5)$$

$$\square$$

### 1.4 Proof of Theorem 4

We start by proving a lemma:

**Lemma 2.** *For $p \to \infty$,*

$$\lambda^\star \to \left( 1 + \frac{\sum_i \mathbb{E}^2 \left[ \Sigma_{ii} - T_{ii} \right]}{\sum_{i,j} \mathrm{Var}\left( S_{ij}' \right)} \right)^{-1}. \tag{6}$$

*Proof.* For this proof, we start by manipulating the analytic formula for $\lambda^*$ in the eigenbasis:

$$\lambda^\star = \frac{\sum_{i,j} \operatorname{Var}\left(S'_{ij}\right) - \sum_i \operatorname{Cov}\left(S'_{ii}, T_{ii}\right)}{\sum_{i,j} \operatorname{Var}\left(S'_{ij}\right) + \sum_i \left\{\operatorname{Var}\left(T_{ii}\right) + \mathbb{E}^2\left[\Sigma_{ii} - T_{ii}\right]\right\}}$$

$$= \left(1 - \frac{\sum_i \operatorname{Cov}\left(S'_{ii}, T_{ii}\right)}{\sum_{i,j} \operatorname{Var}\left(S'_{ij}\right)}\right) \cdot \left(1 + \frac{\sum_i \operatorname{Var}\left(T_{ii}\right)}{\sum_{i,j} \operatorname{Var}\left(S'_{ij}\right)} + \frac{\sum_i \mathbb{E}^2\left[\Sigma_{ii} - T_{ii}\right]}{\sum_{i,j} \operatorname{Var}\left(S'_{ij}\right)}\right)^{-1}.$$

Now, in order to proof the lemma, we have to show that

$$\lim_{p \to \infty} \frac{\sum_i \operatorname{Cov}\left(S'_{ii}, T_{ii}\right)}{\sum_{i,j} \operatorname{Var}\left(S'_{ij}\right)} = 0 \qquad \text{and} \qquad \lim_{p \to \infty} \frac{\sum_i \operatorname{Var}\left(T_{ii}\right)}{\sum_{i,j} \operatorname{Var}\left(S'_{ij}\right)} = 0.$$

hold. We first show that

$$\sum_i \operatorname{Cov}\left(S'_{ii}, T_{ii}\right) = \sum_i \operatorname{Var}\left(T_{ii}\right) = \frac{1}{pn} \sum_{ij} \operatorname{Cov}\left(y_{i1}^2, y_{j1}^2\right).$$

For the covariance expression, we have

$$\sum_{ij} \operatorname{Cov}\left(S'_{ij}, T_{ij}\right) = \sum_i \operatorname{Cov}\left(S'_{ii}, T_{ii}\right)$$

$$= \sum_i \operatorname{Cov}\left(\frac{1}{n}\sum_s y_{is}^2, \frac{1}{np}\sum_{jt} y_{jt}^2\right)$$

$$= \frac{1}{pn^2} \sum_{ij} \sum_{st} \operatorname{Cov}\left(y_{is}^2, y_{jt}^2\right)$$

$$= \frac{1}{pn} \sum_{ij} \operatorname{Cov}\left(y_{i1}^2, y_{j1}^2\right).$$

For the variance of the $T_{ij}$, we have

$$\sum_{ij} \operatorname{Var}\left(T_{ij}\right) = \sum_i \operatorname{Var}\left(T_{ii}\right) = \sum_i \operatorname{Var}\left(\frac{1}{np}\sum_{jt} y_{jt}^2\right)$$

$$= \frac{1}{p^2 n^2} \sum_{it} \operatorname{Var}\left(\sum_j y_{jt}^2\right)$$

$$= \frac{1}{pn} \operatorname{Var}\left(\sum_j y_{j1}^2\right)$$

$$= \frac{1}{pn} \sum_{ij} \operatorname{Cov}\left(y_{i1}^2, y_{j1}^2\right).$$

In the next step, we show that $\sum_{ij} \operatorname{Var}\left(S'_{ij}\right) \geq \frac{1}{n} \sum_{ij} \operatorname{Cov}\left(y_{i1}^2, y_{j1}^2\right)$:

$$\sum_{ij} \operatorname{Var}\left(S'_{ij}\right) = \frac{1}{n^2} \sum_{ij} \operatorname{Var}\left(\sum_t y_{it} y_{jt}\right)$$

$$= \frac{1}{n} \sum_{ij} \operatorname{Var}\left(y_{i1} y_{j1}\right)$$

$$= \frac{1}{n} \sum_{ij} \left\{\mathbb{E}\left[y_{i1}^2 y_{j1}^2\right] - \mathbb{E}^2\left[y_{i1} y_{j1}\right]\right\}$$

$$= \frac{1}{n} \sum_{ij} \left\{\operatorname{Cov}\left(y_{i1}^2, y_{j1}^2\right) + \mathbb{E}\left[y_{i1}^2\right] \mathbb{E}\left[y_{j1}^2\right] - \mathbb{E}^2\left[y_{i1} y_{j1}\right]\right\}$$

$$\geq \frac{1}{n} \sum_{ij} \mathrm{Cov}\left(y_{i1}^2, y_{j1}^2\right) = p \sum_{ij} \mathrm{Var}\left(T_{ij}\right).$$

Putting things together, we have

$$\lim_{p \to \infty} \frac{\sum_i \mathrm{Cov}\left(S'_{ii}, T_{ii}\right)}{\sum_{i,j} \mathrm{Var}\left(S'_{ij}\right)} = \lim_{p \to \infty} \frac{\sum_i \mathrm{Var}\left(T_{ii}\right)}{\sum_{ij} \mathrm{Var}\left(S'_{ij}\right)} \leq \lim_{p \to \infty} \frac{1}{p} = 0$$

and eq. (6) follows. $\square$

With Lemma 2, we can now prove Theorem 4:

**Theorem 4** (Kolmogorov Limit behaviour)**.**

*Proof.* In order to prove the statements, we have to find bounds for the variance term $\sum_{i,j} \mathrm{Var}\left(S'_{ij}\right)$ and the eigenvalue dispersion $\sum_i \mathbb{E}^2\left[\Sigma_{ii} - T_{ii}\right]$. The bounds on the dispersion are nearly directly given by **A4′**, because of the fast convergence of the estimate of the average eigenvalue $T_{ii} \to p^{-1} \sum_i \gamma_i$:

$$\lim_{p \to \infty} \sum_i \mathbb{E}^2\left[\Sigma_{ii} - T_{ii}\right] = \lim_{p \to \infty} \sum_i (\gamma_i - \frac{1}{p}\sum_j \gamma_j)^2 = \Theta\left(\max\left(p, p^{2k}\right)\right).$$

The variance term we analyse in the eigenbasis. For the lower bound, we distinguish two cases: for $k = 1$, we have

$$\sum_{i,j} \mathrm{Var}\left(S'_{ij}\right) \geq \sum_i \mathrm{Var}\left(S'_{ii}\right)$$

$$= \frac{1}{n} \sum_i \left\{\mathbb{E}\left[y_{i1}^4\right] - \mathbb{E}^2\left[y_{i1}^2\right]\right\}$$

$$\geq \frac{1}{n} \sum_i \beta_4 \mathbb{E}^2\left[y_{i1}^2\right] = \frac{\beta_4 p}{n}\frac{1}{p}\sum_i \gamma_i^2$$

$$= \Theta\left(\max\left(1, p^{2k-1}\right)\right) = \Theta(p).$$

For the case $k < 1$, we have

$$\sum_{i,j} \mathrm{Var}\left(S'_{ij}\right) = \frac{1}{n} \sum_{i,j} \left\{\mathbb{E}\left[y_{i1}^2 y_{j1}^2\right] - \mathbb{E}^2\left[y_{i1}y_{j1}\right]\right\}$$

$$\geq \frac{1}{n} \sum_{i,j} \left\{\mathbb{E}\left[y_{i1}^2\right]\mathbb{E}\left[y_{j1}^2\right] - \mathbb{E}^2\left[y_{i1}y_{j1}\right]\right\}$$

$$\geq \frac{1}{n}\left(\sum_i \mathbb{E}\left[y_{i1}^2\right]\right)^2 - \frac{1}{n}\sum_i \mathbb{E}^2\left[y_{i1}^2\right]$$

$$\geq \frac{p^2}{n}\left(\frac{1}{p}\sum_i \mathbb{E}\left[x_{i1}^2\right]\right)^2 - \frac{p}{n}\frac{1}{p}\sum_i \gamma_i^2$$

$$= \Theta(p) - \Theta(\max(1, p^{2k-1})) = \Theta(p).$$

For the upper bound, we have

$$\sum_{i,j} \mathrm{Var}\left(S'_{ij}\right) \leq \frac{1}{n}\sum_{i,j}\left\{\sqrt{\mathrm{Var}(y_{i1}^2)\mathrm{Var}(y_{j1}^2)} + \mathbb{E}\left[y_{i1}^2\right]\mathbb{E}\left[y_{j1}^2\right] - \mathbb{E}^2\left[y_{i1}y_{j1}\right]\right\}$$

$$\leq \frac{2}{n}\sum_{i,j}\sqrt{\mathbb{E}[y_{i1}^4]\mathbb{E}[y_{j1}^4]} \leq \frac{2p^2}{n}(1+\alpha_4)\left(\frac{1}{p}\sum_i \mathbb{E}[y_{i1}^2]\right)^2 = \Theta(p).$$

As the lower and the upper bound are identical, we have

$$\lim_{p \to \infty} \sum_{i,j} \mathrm{Var}\left(S'_{ij}\right) = \Theta(p).$$

Comparing with the dispersion, we we see that

- for $k \leq 0.5$ both terms grow at the same rate, therefore $\lambda^*$ neither goes to zero nor goes to 1.

- for $k > 0.5$, the dispersion grows with rate $\Theta(p^{2k})$, faster than the rate of the variance. Hence $\lambda^*$ goes to zero.

$\square$

## 1.5 Proof of Theorem 5

Before we can prove Theorem 5, we have to obtain a rate on the convergence of the eigendecomposition.

**Lemma 3** (convergence of eigendecomposition)**.**

*Proof.* We follow the same steps as in [2]. Let us define

$$s_u^2 := \frac{1}{n} \sum_{\nu=1}^{n} (u^\nu)^2, \qquad \kappa := \|\mathbf{v}\| s_u,$$

$$\rho_j := \frac{1}{n s_u} \sum_{\nu=1}^{n} u^\nu \xi_j^\nu, \qquad \beta_{ij} := \frac{1}{n} \sum_{\nu=1}^{n} \xi_i^\nu \xi_j^\nu.$$

With these definitions, let us write down the sample covariance matrix in a basis where $\mathbf{v}$ is collinear to the first basis vector:

$$S_n = \begin{pmatrix} \kappa^2 + 2\kappa\rho_1 + \beta_{11} & b_2 & \dots & b_p \\ b_2 & \beta_{22} & \dots & \beta_{2p} \\ \vdots & \vdots & \ddots & \vdots \\ b_p & \beta_{p2} & \dots & \beta_{pp} \end{pmatrix},$$

where $b_i = \kappa\rho_i + \beta_{1j}$. Let us now rotate in the eigenbasis of the $\beta$-submatrix:

$$V S_n V^{\mathsf{T}} = \begin{pmatrix} \kappa^2 + 2\kappa\rho_1 + \beta_{11} & \tilde{b}_2 & \dots & \tilde{b}_p \\ \tilde{b}_2 & \lambda_2 & \dots & 0 \\ \vdots & \vdots & \ddots & \vdots \\ \tilde{b}_p & 0 & \dots & \lambda_p \end{pmatrix} \tag{7}$$

where $\tilde{b}_i = \kappa\tilde{\rho}_i + \tilde{\beta}_{1j}$ with

$$\tilde{\rho}_j = \frac{1}{n s_u} \sum_{\nu=1}^{n} u^\nu \tilde{\xi}_j^\nu, \qquad \tilde{\beta}_{ij} = \frac{1}{n} \sum_{\nu=1}^{n} \tilde{\xi}_i^\nu \tilde{\xi}_j^\nu.$$

where $\tilde{\xi}_i^\nu$ is the noise in the $\beta$-eigenbasis. Matrices like the one in eq. (7) are called arrowhead matrices, their characteristic equations have a simple form:

$$f(\lambda) = (\lambda - \kappa^2 - 2\kappa\rho_1 - \beta_{11}) - \sum_{j=2}^{p} \frac{\tilde{b}_j^2}{\lambda - \lambda_j}. \tag{8}$$

To solve the characteristic equation, we use the Marchenko-Pastur law which yields the density of the eigenvalues $\lambda_j$ of the submatrix:

$$f_{\mathrm{MP}}(x) = \frac{1}{2\pi x c} \sqrt{(b - x)(x - a)}, \qquad x \in [a, b],$$

with $a = (1 - \sqrt{c})^2$ and with $b = (1 + \sqrt{c})^2$.

As the random variable $\tilde{b}_j$ has variance $\lambda_j(\kappa^2 + 2\kappa\rho_1 + \beta_{11})/n$, we introduce a new random variable $\eta_j = (\lambda_j(\kappa^2 + 2\kappa\rho_1 + \beta_{11})/n)^{-1/2} \tilde{b}_j$. We obtain

$$\sum_{j=2}^{p} \frac{\tilde{b}_j^2}{\lambda - \lambda_j} = (\kappa^2 + 2\kappa\rho_1 + \beta_{11}) \frac{p-1}{n} \frac{1}{p-1} \sum_{j=2}^{p} \frac{\lambda_j n_j^2}{\lambda - \lambda_j}.$$

In the large $n$, $p$ limit we have $\kappa^2 \to 1$, $\rho_1 = \mathcal{O}(1/\sqrt{p}) \to 0$ and $\beta_{11} \to \mathrm{Var}(\beta_{11}) = 1$. The fluctuations in the $\eta_j^2$ average out – therefore, we have

$$\lim_{p,n,\to\infty} \sum_{j=2}^{p} \frac{\tilde{b}_j^2}{\lambda - \lambda_j} = (\|\mathbf{v}\|^2 + 1)\frac{p-1}{n} \int_a^b f_{\mathrm{MP}}(x)\frac{x}{\lambda - x}dx.$$

Plugging this into eq. (8) and solving for $\lambda$, we obtain

$$\lambda(\alpha) = \alpha + c\frac{\alpha}{\alpha - 1}.$$

For the eigenvectors of the arrowhead matrix, there is also a closed form expression: up to normalization, the eigenvector corresponding to the eigenvalue $\lambda$ is given by

$$\mathbf{v} = \left(1, \frac{\tilde{b}_2}{\lambda - \lambda_2}, \dots, \frac{\tilde{b}_p}{\lambda - \lambda_p}\right)$$

and the overlap is therefore

$$R^2 = \frac{\langle \mathbf{v}, \mathbf{e}_1 \rangle}{\|\mathbf{v}\|^2} = \frac{1}{\sum_{j=2}^{p} \tilde{b}_j^2/(\lambda - \lambda_j)^2}.$$

In the large $n$, $p$ limit this again turns into an integral,

$$\lim_{p,n,\to\infty} R^2 = \frac{1}{1 + p/n \int_a^b f_{\mathrm{MP}}(x)x/(\lambda - x)^2 dx}.$$

whose evaluation finishes the proof. $\qquad\square$

Before we start with the proof of the optimality of *oc-Shrinkage*, let us introduce some notation. Let $\mathbf{Y}$ denote data in their eigenbasis and $e_i$ denote the sample eigenvectors, sorted by decreasing eigenvalue. Based on the $e_i$, we define a new orthonormal set of vectors $v_i$ such that:

$$v_1 = e_1 \qquad \text{and} \qquad \langle v_i, e_j \rangle = 0 \quad \text{if} \quad j \neq i, j \neq 1. \tag{9}$$

For the rotated data set $U = VY$ we then have

$$u_{it} = y_{it}\langle v_i, e_i \rangle + y_{1t}\langle v_i, e_1 \rangle := y_{it}\sqrt{1 - \varepsilon_i^2} + y_{1t}\varepsilon_i, \tag{10}$$

making the covariance matrix an arrowhead matrix [2]. The $\varepsilon_i$ quantify the error in the estimate of the first eigendirection: if all $\varepsilon_i$ are zero, the first eigendirection and the first sample eigendirection coincide. We have the following lemma:

**Lemma 4.** *In the limit, for the sum over the squared $\varepsilon_i$, we have*

$$\lim_{p\to\infty} \sum_{i\geq 2} \varepsilon_i^2 \leq K_6 n^{-k}.$$

*Proof.* By definition, we have

$$\lim_{p\to\infty} \sum_{i\geq 2} \varepsilon_i^2 = \lim_{p\to\infty} \sum_{i\geq 2} |\langle e_1, v_i \rangle|^2$$

$$= \lim_{p\to\infty} \left(1 - |\langle e_1, v_1 \rangle|^2\right).$$

Using Theorem 3 –here the noise is even bounded by a constant– we have

$$\leq \lim_{p\to\infty} \left(1 - \frac{np^{2k-1} - 1}{np^{2k-1} + p^k}\right)$$

$$\leq \lim_{p\to\infty} \left(\frac{p^k}{np^{2k-1} + p^k}\right)$$

$$\leq \lim_{p\to\infty} \left(\frac{1}{np^{k-1}+1}\right)$$

$$\leq \lim_{p\to\infty} K_6 n^{-k}.$$

$\square$

**Corollary 1.** *In the limit, for the sum over the absolute values of the $\varepsilon_i$,*

$$\lim_{p\to\infty} \sum_{i\geq 2} |\varepsilon_i| \leq \sqrt{K_6} n^{1/2-k/2}$$

*holds.*

*Proof.* Let us assume

$$\sum_{i\geq 2} \varepsilon_i^2 = K_6 n^{-k}.$$

Then, the sum over the absolute values of $\varepsilon_i$ is maximized if $\varepsilon_i = \varepsilon$ holds. Then, we have

$$\sum_{i\geq 2} \varepsilon^2 = p\varepsilon^2 = K_6 n^{-k} \Leftrightarrow \varepsilon = \pm\sqrt{K_6} n^{-k/2-1/2}$$

$$\Rightarrow \sum_{i\geq 2} |\varepsilon_i| \leq \sqrt{K_6} n^{1/2-k/2}.$$

$\square$

For Theorems 5 and **??** we need to know under which circumstances we are allowed to exchange maximization and summation:

**Lemma 5.** *Let* **A1$'$** *hold and $X_1, ..., X_p$ be a set of random variables, $\mu_i := \mathbb{E}[X_i]$, $\sigma_i^2 := \mathbb{E}[X_i^2]$ and $\mathbb{E}[X_i^4] < \infty$, and let $x_{in}$ denote the $n^{th}$ realization of the $i^{th}$ random variable. Then,*

$$\lim_{p\to\infty} \max_i \frac{1}{n} \sum_m x_{im} = \max_i \mathbb{E}[X_i].$$

*Proof.* Let $\mu_{max} := \max_i \mu_i$. The probability for the maximum of the l.h.s being larger than the r.h.s. plus $\epsilon$ is

$$\lim_{p\to\infty} \sum_i P\left(\frac{1}{n}\sum_m x_{im} > \max_i \mathbb{E}[X_i] + \epsilon\right)$$

$$= \lim_{p\to\infty} \sum_i P\left(\frac{1}{n}\sum_m x_{im} - \mu_i > \max_i \mu_i - \mu_i + \epsilon\right)$$

$$\leq \lim_{p\to\infty} \sum_i P\left(\frac{1}{n}\sum_m (x_{im} - \mu_i) > \epsilon\right)$$

to show that this probability goes to zero, we use the following inequality:

$$P(y > \epsilon)\epsilon^4 = \int_\epsilon^\infty dy\, p(y)\epsilon^4 \leq \int_\epsilon^\infty dy\, p(y)y^4 \leq \mathbb{E}[y^4]$$

$$\leq \lim_{p\to\infty} \sum_i \frac{1}{\epsilon^4}\mathbb{E}\left[\left\{\frac{1}{n}\sum_m (x_{im} - \mu_i)\right\}^4\right]$$

$$\leq \lim_{p\to\infty} \sum_i \frac{1}{\epsilon^4}\left\{\frac{1}{n^3}\mathbb{E}\left[(x_{in}-\mu_i)^4\right] + \frac{3(n-1)}{n^3}\mathbb{E}^2\left[(x_{in}-\mu_i)^2\right]\right\}$$

$$\leq \lim_{p\to\infty} \frac{K_1}{\epsilon^4}\left\{\frac{1}{n^2}\max_i \mathbb{E}\left[(x_{in}-\mu_i)^4\right] + \frac{3(n-1)}{n^2}\max_i \mathbb{E}^2\left[(x_{in}-\mu_i)^2\right]\right\}$$

$$= 0.$$

□

Now we are ready to prove that, asymptotically, our procedure obtains the optimal shrinkage strength $\lambda_{\widehat{oc}}^*$ on the estimated orthogonal complement:

**Theorem 5** (consistency of oc-shrinkage)**.**

*Proof.* The normalized mean squared error on the sample orthogonal complement is defined by

$$MSE_{\widehat{oc}}(\lambda) := \frac{1}{p} \sum_{i,j \geq 2} \{C_{\widehat{oc}} - (1-\lambda)S_{\widehat{oc}} - \lambda T_{\widehat{oc}}\}_{ij}^2$$

$$= \frac{1}{p} \sum_{i,j \geq 2} (C_{\widehat{oc},ij} - S_{\widehat{oc},ij})^2 + \lambda^2 (S_{\widehat{oc},ij} - T_{\widehat{oc},ij})^2 + 2\lambda(C_{\widehat{oc},ij} - S_{\widehat{oc},ij})(S_{\widehat{oc},ij} - T_{\widehat{oc},ij}).$$

Minimization yields

$$\arg\min_\lambda MSE_{\widehat{oc}}(\lambda) = \frac{\frac{1}{p}\sum_{i,j\geq 2}(C_{\widehat{oc},ij} - S_{\widehat{oc},ij})(S_{\widehat{oc},ij} - T_{\widehat{oc},ij})}{\frac{1}{p}\sum_{i,j\geq 2}(S_{\widehat{oc},ij} - T_{\widehat{oc},ij})^2} := \frac{W_{\widehat{oc}}}{\hat{E}_{\widehat{oc}}}.$$

We have to show that, in the limit, this is equal to

$$\hat{\lambda}_{\widehat{oc}} = \frac{\hat{V}_{\widehat{oc}}}{\hat{E}_{\widehat{oc}}}.$$

As the denominator is the same, we have to show that the numerator is identical in the limit. Therefore, we prove the following equalities:

$$\lim_{p\to\infty} W_{\widehat{oc}} \overset{(1)}{=} \lim_{p\to\infty} W_{oc} \overset{(2)}{=} \lim_{p\to\infty} V_{oc} \overset{(3)}{=} \lim_{p\to\infty} \hat{V}_{oc} \overset{(4)}{=} \lim_{p\to\infty} \hat{V}_{\widehat{oc}}. \tag{11}$$

Equalities (1) and (4) state that, in the limit, it does not matter if the quantities are calculated on the true or on the sample orthogonal complement. On the true orthogonal complement, we can prove the desired equality.

Equality (1) is proven in Lemma 6, equality (2) in Lemma 7. By the consistency of Shrinkage, equality (3) holds. Last, equality (4) is proven in Lemma 8.

For Lemmas 6, 7 and 8, see Section 1.6. □

## 1.6 Additional lemmas (Lemma 6-8)

For the following lemmas, we use the rotated version of the data defined in eq. (10), using the basis from eq. (9).

**Lemma 6.** *Under the above assumptions,*

$$\lim_{p\to\infty} W_{oc} = \lim_{p\to\infty} W_{\widehat{oc}}.$$

*Proof of Lemma 6.* We have

$$W_{\widehat{oc}} = \sum_{i,j\geq 2} \frac{1}{p}(C_{\widehat{oc},ij} - S_{\widehat{oc},ij})(S_{\widehat{oc},ij} - T_{\widehat{oc},ij})$$

$$= p^{-1} \sum_{i,j\geq 2} C_{\widehat{oc},ij}S_{\widehat{oc},ij} - C_{\widehat{oc},ij}T_{\widehat{oc},ij} - S_{\widehat{oc},ij}^2 + S_{\widehat{oc},ij}T_{\widehat{oc},ij}. \tag{12}$$

Now we have to show that, in the limit, the four terms are equal to those in the true orthogonal complement. Let us start with the first term in eq. (12):

$$\lim_{p\to\infty} p^{-1} \sum_{i,j\geq 2} C_{\widehat{oc},ij}S_{\widehat{oc},ij} = \lim_{p\to\infty} (pn)^{-1} \sum_{i,j\geq 2} \mathbb{E}[u_i u_j] \sum_s u_{is}u_{js}$$

$$= \lim_{p \to \infty} (pn)^{-1} \sum_{i,j \geq 2} \mathbb{E}\left[\left(y_i\sqrt{1-\varepsilon_i^2} + y_1\varepsilon_i\right)\left(y_j\sqrt{1-\varepsilon_j^2} + y_1\varepsilon_j\right)\right]$$

$$\cdot \sum_s \left(y_i\sqrt{1-\varepsilon_i^2} + y_1\varepsilon_i\right)\left(y_j\sqrt{1-\varepsilon_j^2} + y_1\varepsilon_j\right)$$

$$= \lim_{p \to \infty} (pn)^{-1} \sum_{i,j \geq 2} \sum_s \left\{ \mathbb{E}\left[y_iy_j\right] y_{is}y_{js} \overset{(2)}{+} \mathbb{E}\left[y_iy_j\right] y_{is}y_{js}(\varepsilon_i^2\varepsilon_j^2 - \varepsilon_i^2 - \varepsilon_j^2) \right.$$

$$\overset{(3)}{+} \mathbb{E}\left[y_1^2\right] y_{1s}^2\varepsilon_i^2\varepsilon_j^2$$

$$\overset{(4)}{+} 2\mathbb{E}\left[y_iy_1\right] y_{is}y_{1s}(1-\varepsilon_i^2)\varepsilon_j^2$$

$$\overset{(5)}{+} \left(2\mathbb{E}\left[y_iy_1\right] y_{js}y_{1s} + \mathbb{E}\left[y_iy_j\right] y_{1s}y_{1s} + \mathbb{E}\left[y_1y_1\right] y_{is}y_{js}\right)\sqrt{(1-\varepsilon_i^2)}\sqrt{(1-\varepsilon_j^2)}\varepsilon_i\varepsilon_j$$

$$\overset{(6)}{+} 2\mathbb{E}\left[y_iy_j\right] y_{is}y_{1s}(1-\varepsilon_i^2)\sqrt{(1-\varepsilon_j^2)}\varepsilon_j$$

$$\overset{(7)}{+} 2\mathbb{E}\left[y_iy_1\right] y_{1s}^2\sqrt{(1-\varepsilon_i^2)}\varepsilon_i\varepsilon_j^2$$

$$\overset{(8)}{+} 2\mathbb{E}\left[y_1y_i\right] y_{is}y_{js}(1-\varepsilon_i^2)\sqrt{(1-\varepsilon_j^2)}\varepsilon_j$$

$$\left. \overset{(9)}{+} 2\mathbb{E}\left[y_1^2\right] y_{is}y_{1s}\sqrt{(1-\varepsilon_i^2)}\varepsilon_i\varepsilon_j^2 \right\}.$$

The first term is equal to $C_{oc,ij}S_{oc,ij}$, we therefore have to show that the terms (2)-(9) go to zero. As the covariance matrix of the $y_i$ is diagonal, the terms (4), (7) and (8) are always zero. Let us start with term (2):

$$\lim_{p \to \infty} (pn)^{-1} \left| \sum_{i,j \geq 2} \sum_s \mathbb{E}\left[y_iy_j\right] y_{is}y_{js}(\varepsilon_i^2\varepsilon_j^2 - \varepsilon_i^2 - \varepsilon_j^2) \right|$$

$$= \lim_{p \to \infty} (pn)^{-1} \sum_{i \geq 2} \mathbb{E}\left[y_i^2\right] \left|\varepsilon_i^4 - 2\varepsilon_i^2\right| \left|\sum_s y_{is}^2\right|$$

$$\leq \lim_{p \to \infty} 2(pn)^{-1} \sum_{i \geq 2} \mathbb{E}\left[y_i^2\right] \varepsilon_i^2 \left|\sum_s y_{is}^2\right|$$

$$= \lim_{p \to \infty} 2(pn)^{-1} K_6 n^{-k} \max_{i \geq 2} \mathbb{E}\left[y_i^2\right] \left|\sum_s y_{is}^2\right|$$

$$= \lim_{p \to \infty} 2p^{-1} K_6 n^{-k} \max_{i \geq 2} \mathbb{E}\left[y_i^2\right] \mathbb{E}[y_i^2]$$

$$= 0.$$

For term (3), we have

$$\lim_{p \to \infty} (pn)^{-1} \left| \sum_{i,j \geq 2} \sum_s \mathbb{E}\left[y_1^2\right] y_{1s}^2\varepsilon_i^2\varepsilon_j^2 \right|$$

$$\leq \lim_{p \to \infty} (pn)^{-1} K_6^2 n^{-2k} \mathbb{E}\left[y_1^2\right] \sum_s y_{1s}^2$$

$$= \lim_{p \to \infty} (pn)^{-1} K_6^2 n^{-2k} n\mathbb{E}^2\left[y_1^2\right]$$

$$= \lim_{p \to \infty} p^{-1} K_6^2 \mathbb{E}^2\left[z_1^2\right]$$

$$= 0.$$

Term (5) is

$$\lim_{p\to\infty} \frac{1}{pn} \left| \sum_{i,j\geq2} \sum_s \left(2\mathbb{E}\left[y_iy_1\right]y_{js}y_{1s} + \mathbb{E}\left[y_iy_j\right]y_{1s}^2 + \mathbb{E}\left[y_1^2\right]y_{is}y_{js}\right)\sqrt{(1-\varepsilon_i^2)}\sqrt{(1-\varepsilon_j^2)}\varepsilon_i\varepsilon_j \right|$$

$$\leq \lim_{p\to\infty} \frac{1}{pn} \sum_{i,j\geq2} \left|\sqrt{(1-\varepsilon_i^2)}\sqrt{(1-\varepsilon_j^2)}\varepsilon_i\varepsilon_j\right| \left|\sum_s \left(\mathbb{E}\left[y_iy_j\right]y_{1s}^2 + \mathbb{E}\left[y_1^2\right]y_{is}y_{js}\right)\right|$$

$$\leq \lim_{p\to\infty} \frac{1}{pn} \sum_{i\geq2} \varepsilon_i^2 \left|\sum_s \mathbb{E}\left[y_i^2\right]y_{1s}^2\right| + \sum_{i,j\geq2} \left|\varepsilon_i\varepsilon_j\right| \left|\sum_s \mathbb{E}\left[y_1^2\right]y_{is}y_{js}\right|$$

$$\leq \lim_{p\to\infty} \frac{1}{p} \left\{ K_6 n^{-k}\mathbb{E}[y_1^2] \max_{i\geq2}\mathbb{E}\left[y_i^2\right] + \mathbb{E}\left[y_1^2\right]n^{-1}\sum_{i\geq2}\varepsilon_i^2 \left|\sum_s y_{is}^2\right| + \mathbb{E}\left[y_1^2\right]n^{-1}\sum_{i,j\geq2,j\neq i}\left|\varepsilon_i\varepsilon_j\right|\left|\sum_s y_{is}y_{js}\right| \right\}$$

$$\leq \lim_{p\to\infty} \frac{1}{p} \left\{ K_6 n^{-k}\left(\mathbb{E}[y_1^2]\max_{i\geq2}\mathbb{E}\left[y_i^2\right] + \mathbb{E}\left[y_1^2\right]\max_{i\geq2}\mathbb{E}[y_i^2]\right) + 2\mathbb{E}\left[y_1^2\right]n^{-1}\sum_{i,j\geq2,j\neq i}\varepsilon_i^2\left|\sum_s y_{is}y_{js}\right| \right\}$$

$$\leq \lim_{p\to\infty} \frac{2K_6}{p} \left\{ \mathbb{E}[z_1^2]\max_{i\geq2}\mathbb{E}\left[z_i^2\right] + \mathbb{E}\left[z_1^2\right]\max_{i\geq2}\sum_{j\geq2,j\neq i}\mathbb{E}[z_iz_j] \right\}$$

$$= 0.$$

For the term (6), we have

$$\lim_{p\to\infty} (pn)^{-1} \left| \sum_{i,j\geq2}\sum_s 2\mathbb{E}\left[y_iy_j\right]y_{is}y_{1s}(1-\varepsilon_i^2)\sqrt{(1-\varepsilon_j^2)}\varepsilon_j \right|$$

$$\leq \lim_{p\to\infty} 2p^{-1}\sum_{i\geq2}\left|\varepsilon_i\right|\left|n^{-1}\sum_s\mathbb{E}\left[y_i^2\right]y_{is}y_{1s}\right|$$

$$\leq \lim_{p\to\infty} 2p^{-1}n^{1/2}\max_{i\geq2}\left|n^{-1}\sum_s\mathbb{E}\left[z_i^2\right]z_{is}z_{1s}\right|$$

$$\leq \lim_{p\to\infty} 2p^{-1}n^{1/2}\max_{i\geq2}\left|\mathbb{E}\left[z_i^2\right]\mathbb{E}[z_iz_1]\right|$$

$$= 0.$$

Finally, we have term (9)

$$\lim_{p\to\infty} (pn)^{-1} \left| \sum_{i,j\geq2}\sum_s 2\mathbb{E}\left[y_1^2\right]y_{is}y_{1s}\sqrt{(1-\varepsilon_i^2)}\varepsilon_i\varepsilon_j^2 \right|$$

$$\leq \lim_{p\to\infty} 2p^{-1}K_6 n^{k/2}\mathbb{E}\left[z_1^2\right]\sum_{i\geq2}\left|\varepsilon_i\right|\left|n^{-1}\sum_s z_{is}z_{1s}\right|$$

$$\leq \lim_{p\to\infty} 2p^{-1}K_6 n^{k/2}n^{1/2}\mathbb{E}\left[z_1^2\right]\max_{i\geq2}\left|n^{-1}\sum_s z_{is}z_{1s}\right|$$

$$= \lim_{p\to\infty} 2p^{-1}K_6 n^{k/2}n^{1/2}\mathbb{E}\left[z_1^2\right]\max_{i\geq2}\left|\mathbb{E}[z_iz_1]\right|$$

$$= 0.$$

Let us continue with the second term in eq. (12):

$$\lim_{p\to\infty} p^{-1}\sum_{i,j\geq2} C_{\widehat{oc},ij}T_{\widehat{oc},ij} = \lim_{p\to\infty} p^{-1}\sum_{i\geq2}C_{\widehat{oc},ii}T_{\widehat{oc},ii}$$

$$= \lim_{p\to\infty} p^{-2}n^{-1}\sum_{i\geq 2}\mathbb{E}\left[\left(y_i\sqrt{1-\varepsilon_i^2}+y_1\varepsilon_i\right)^2\right]\sum_{k\geq 2}\sum_s\left(y_{ks}\sqrt{1-\varepsilon_k^2}+y_{1s}\varepsilon_k\right)^2$$

$$= \lim_{p\to\infty} p^{-2}n^{-1}\sum_{i\geq 2}\mathbb{E}\left[\left(y_i^2(1-\varepsilon_i^2)+y_1^2\varepsilon_i^2\right)\right]\sum_{k\geq 2}\sum_s\left(y_{ks}^2(1-\varepsilon_k^2)+y_{1s}^2\varepsilon_k^2+2y_{ks}y_{1s}\sqrt{1-\varepsilon_k^2}\varepsilon_k\right)$$

$$= \lim_{p\to\infty} p^{-2}n^{-1}\Bigg\{\sum_{i\geq 2}\mathbb{E}\left[y_i^2\right]\sum_{k\geq 2}\sum_s y_{ks}^2$$

$$+\sum_{i\geq 2}\mathbb{E}\left[-y_i^2\varepsilon_i^2+y_1^2\varepsilon_i^2\right]\sum_{k\geq 2}\sum_s\left(-y_{ks}^2\varepsilon_k^2+y_{1s}^2\varepsilon_k^2+2y_{ks}y_{1s}\sqrt{1-\varepsilon_k^2}\varepsilon_k\right)\Bigg\}.$$

The first term is equal to $C_{oc,ij}S_{oc,ij}$, we therefore have to show that the remainder goes to zero:

$$\lim_{p\to\infty}\left|p^{-2}n^{-1}\sum_{i\geq 2}\mathbb{E}\left[-y_i^2\varepsilon_i^2+y_1^2\varepsilon_i^2\right]\sum_{k\geq 2}\sum_s\left(-y_{ks}^2\varepsilon_k^2+y_{1s}^2\varepsilon_k^2+2y_{ks}y_{1s}\sqrt{1-\varepsilon_k^2}\varepsilon_k\right)\right|$$

$$\leq \lim_{p\to\infty} p^{-2}K_6 n^{-k}\max_{i\geq 2}\left|\mathbb{E}[y_i^2]+\mathbb{E}[y_1^2]\right|\max_{k\geq 2}\left|n^{-1}\sum_s\left(-y_{ks}^2 K_6 n^{-k}+y_{1s}^2 K_6 n^{-k}+2y_{ks}y_{1s}\sqrt{K_6}n^{-k/2}\right)\right|$$

$$= \lim_{p\to\infty} p^{-2}K_6^2 n^{-2k}\max_{i\geq 2}\left|\mathbb{E}[y_i^2]+\mathbb{E}[y_1^2]\right|\max_{k\geq 2}\left|-\mathbb{E}[y_k^2]+\mathbb{E}[y_1^2]\right|$$

$$= 0.$$

Let us now have a look at the third term in eq. (12):

$$\lim_{p\to\infty} p^{-1}\sum_{i,j\geq 2}S_{\hat{o}c,ij}^2 = \lim_{p\to\infty} p^{-1}n^{-2}\sum_{i,j\geq 2}\sum_s u_{is}u_{js}\sum_t u_{it}u_{jt}$$

$$= \lim_{p\to\infty} p^{-1}n^{-2}\sum_{i,j\geq 2}\sum_{s,t}\left(y_{is}\sqrt{1-\varepsilon_i^2}+y_{1s}\varepsilon_i\right)\left(y_{js}\sqrt{1-\varepsilon_j^2}+y_{1s}\varepsilon_j\right)$$

$$\cdot\left(y_{it}\sqrt{1-\varepsilon_i^2}+y_{1t}\varepsilon_i\right)\left(y_{jt}\sqrt{1-\varepsilon_j^2}+y_{1t}\varepsilon_j\right)$$

$$= \lim_{p\to\infty} p^{-1}n^{-2}\sum_{i,j\geq 2}\sum_{s,t}\Bigg\{\overset{(2)}{y_{is}y_{js}y_{it}y_{jt}}+y_{is}y_{js}y_{it}y_{jt}(\varepsilon_i^2\varepsilon_j^2-\varepsilon_i^2-\varepsilon_j^2)$$

$$\overset{(3)}{+}y_{1t}^2 y_{1s}^2\varepsilon_i^2\varepsilon_j^2$$

$$\overset{(4)}{+}2y_{it}y_{1t}y_{is}y_{1s}(1-\varepsilon_i^2)\varepsilon_j^2$$

$$\overset{(5)}{+}(2y_{it}y_{1t}y_{js}y_{1s}+2y_{1s}y_{1s}y_{it}y_{jt})\sqrt{(1-\varepsilon_i^2)}\sqrt{(1-\varepsilon_j^2)}\varepsilon_i\varepsilon_j$$

$$\overset{(6)}{+}4y_{it}y_{jt}y_{is}y_{1s}(1-\varepsilon_i^2)\sqrt{(1-\varepsilon_j^2)}\varepsilon_j$$

$$\overset{(7)}{+}4y_{it}y_{1t}y_{1s}^2\sqrt{(1-\varepsilon_i^2)}\varepsilon_i\varepsilon_j^2\Bigg\}.$$

The first term is equal to $S_{oc,ij}^2$, we therefore have to show that the terms (2)-(9) go to zero. Let us start with term (2):

$$\lim_{p\to\infty}\left|\frac{1}{pn^2}\sum_{i\geq 2,j\geq 2}\sum_{s,t}y_{is}y_{it}y_{js}y_{jt}(\varepsilon_i^2\varepsilon_j^2-\varepsilon_i^2-\varepsilon_j^2)\right|$$

$$\leq \lim_{p\to\infty}\frac{2}{pn^2}\sum_{i\geq 2,j\geq 2}\left|\sum_{s,t}y_{is}y_{it}y_{js}y_{jt}\varepsilon_i^2\right|$$

$$\leq \lim_{p\to\infty} \frac{2}{pn^2} \sum_{i\geq 2} \varepsilon_i^2 \left| \sum_{s,t,j\geq 2} y_{is}y_{it}y_{js}y_{jt} \right|$$

$$\leq \lim_{p\to\infty} \frac{2}{pn^2} K_6 n^{-k} \max_{i\geq 2} \left| \sum_{s,t,j\geq 2} y_{is}y_{it}y_{js}y_{jt} \right|$$

$$j = i : \sum_s y_{is}^4 + \sum_{s,t\neq s} y_{is}^2 y_{it}^2$$

$$j \neq i : \sum_{s,j\geq 2} y_{is}^2 y_{js}^2 + \sum_{s,t,j\geq 2} y_{is}y_{it}y_{js}y_{jt}$$

$$\leq \lim_{p\to\infty} \frac{2K_6}{n^{1+k}} \max_{i\geq 2} \left| \frac{1}{p}\mathbb{E}[y_{i1}^4] + \frac{n-1}{p}\mathbb{E}^2[y_{i1}^2] + \frac{1}{p}\sum_{j\neq i}\mathbb{E}[y_{i1}^2]\mathbb{E}[y_{j1}^2] + \frac{n-1}{p}\sum_{j\neq i}\mathbb{E}^2[y_{i1}y_{j1}] \right|$$

$$= 0.$$

For term (3), we have

$$\lim_{p\to\infty} \left| \frac{1}{pn^2} \sum_{i\geq 2, j\geq 2} \sum_{s,t} y_{1s}^2 y_{1t}^2 \varepsilon_i^2 \varepsilon_j^2 \right|$$

$$\leq \lim_{p\to\infty} \frac{K_6^2 n^{-2k}}{pn^2} \left| \sum_s y_{1s}^4 + \sum_{s,t\neq t} y_{1s}^2 y_{1t}^2 \right|$$

$$\leq \lim_{p\to\infty} \frac{K_6^2 n^{-2k}}{p} \left| \frac{1}{n}\mathbb{E}[y_{1s}^4] + \frac{n-1}{n}\mathbb{E}[y_{1s}^2]\mathbb{E}[y_{1t}^2] \right|$$

$$\leq \lim_{p\to\infty} \frac{K_6^2}{p} \left| \frac{1}{n}\mathbb{E}[z_1^4] + \frac{n-1}{n}\mathbb{E}^2[z_1^2] \right|$$

$$= 0.$$

Term (4) goes to zero as

$$\lim_{p\to\infty} \left| \frac{1}{pn^2} \sum_{i\geq 2, j\geq 2} \sum_{s,t} 2(1-\varepsilon_i^2)\varepsilon_j^2 \cdot y_{is}y_{it}y_{1s}y_{1t} \right|$$

$$\leq \lim_{p\to\infty} \frac{2K_6 n^{-k}}{pn^2} \sum_{i\geq 2} \left| \sum_{s,t} y_{is}y_{it}y_{1s}y_{1t} \right|$$

$$\leq \lim_{p\to\infty} \frac{2K_6 n^{-k}}{p} \sum_{i\geq 2} \left| \frac{1}{n^2}\sum_s y_{is}^2 y_{1s}^2 + \frac{1}{n^2}\sum_{s,t\neq s} y_{is}y_{it}y_{1s}y_{1t} \right|$$

$$\leq \lim_{p\to\infty} \frac{2K_6 n^{-k}}{p} \sum_{i\geq 2} \left| \frac{1}{n}\mathbb{E}[y_{i1}^2 y_{11}^2] + \frac{n-1}{n}\mathbb{E}^2[y_i y_1] \right|$$

$$\leq \lim_{p\to\infty} \frac{2K_6}{p} \sum_{i\geq 2} \left| \frac{1}{n}\mathbb{E}[z_{i1}^2 z_{11}^2] + \frac{n-1}{n}\mathbb{E}^2[z_i z_1] \right|$$

$$= 0.$$

For term (5), we have

$$\lim_{p\to\infty} \left| \frac{1}{pn^2} \sum_{i\geq 2, j\geq 2} \sum_{s,t} 2\varepsilon_i \varepsilon_j \sqrt{1-\varepsilon_i^2}\sqrt{1-\varepsilon_j^2} \cdot (y_{is}y_{1t}y_{jt}y_{1s} + y_{is}y_{js}y_{1t}^2) \right|$$

$$\leq \lim_{p\to\infty} \frac{1}{pn^2} \sum_{i\geq 2, j\geq 2} \varepsilon_i \varepsilon_j \left| \sum_{s,t} 2 \cdot (y_{is}y_{1t}y_{jt}y_{1s} + y_{is}y_{js}y_{1t}^2) \right|$$

$$\leq \lim_{p\to\infty} \frac{2}{pn^2} \sum_{i\geq 2} \varepsilon_i^2 \left| \sum_s (2y_{is}^2 y_{1s}^2) + \sum_{s,t\neq s} (y_{is}y_{1t}y_{it}y_{1s} + y_{is}^2 y_{1t}^2) \right|$$

$$+ \frac{2}{pn^2} \left| \sum_{i\geq 2, j\geq 2\neq i} |\varepsilon_i \varepsilon_j| \sum_s (2y_{is}y_{1s}^2 y_{js}) + \sum_{s,t\neq s} (y_{is}y_{1t}y_{jt}y_{1s} + y_{is}y_{js}y_{1t}^2) \right|$$

$$\leq \lim_{p\to\infty} \frac{2K_6 n^{-k}}{pn^2} \max_{i\geq 2} \left| \sum_s (2y_{is}^2 y_{1s}^2) + \sum_{s,t\neq t} (y_{is}y_{1t}y_{it}y_{1s} + y_{is}^2 y_{1t}^2) \right|$$

$$+ \frac{4K_6 n^{1-k}}{pn^2} \sum_{i\geq 2, j\geq 2\neq i} \varepsilon_i^2 \left| \sum_s (2y_{is}y_{1s}^2 y_{js}) + \sum_{s,t\neq s} (y_{is}y_{1t}y_{jt}y_{1s} + y_{is}y_{js}y_{1t}^2) \right|$$

$$\leq \lim_{p\to\infty} \frac{2K_6}{pn^2} \max_{i\geq 2} \left| \sum_s (2z_{is}^2 z_{1s}^2) + \sum_{s,t\neq s} (z_{is}z_{1t}z_{it}z_{1s} + z_{is}^2 z_{1t}^2) \right|$$

$$+ \frac{2K_6}{pn^2} \max_{i\geq 2} \sum_{j\geq 2\neq i} \left| \sum_s (2z_{is}z_{1s}^2 z_{js}) + \sum_{s,t\neq s} (z_{is}z_{1t}z_{jt}z_{1s} + z_{is}z_{js}z_{1t}^2) \right|$$

$$\leq \lim_{p\to\infty} \frac{2K_6}{p} \max_{i\geq 2} \left| (2n^{-1}\mathbb{E}[z_i^2 z_1^2] + \frac{n-1}{n}(\mathbb{E}^2[z_i z_1] + \mathbb{E}[z_i^2]\mathbb{E}[z_1^2]) \right|$$

$$+ \frac{2K_6}{p} \max_{i\geq 2} \sum_{j\geq 2\neq i} \left| 2n^{-1}\mathbb{E}[z_i z_1^2 z_j] + \frac{n-1}{n}(\mathbb{E}[z_i z_1]\mathbb{E}[z_1 z_j] + \mathbb{E}[z_i z_j]\mathbb{E}[z_1^2]) \right|$$

$$= 0.$$

For term (6), we have

$$\lim_{p\to\infty} \left| \frac{1}{pn^2} \sum_{i\geq 2, j\geq 2} \sum_{s,t} 4y_{it}y_{jt}y_{is}y_{1s}(1-\varepsilon_i^2)\sqrt{(1-\varepsilon_j^2)}\varepsilon_j \right|$$

$$\leq \lim_{p\to\infty} \frac{4}{pn^2} \sum_{i\geq 2, j\geq 2} |\varepsilon_j| \left| \sum_{s,t} y_{it}y_{jt}y_{is}y_{1s} \right|$$

$$\leq \lim_{p\to\infty} \frac{4\sqrt{K_6}n^{-1/2}}{pn^2} \left| \max_{j\geq 2} \sum_{i\geq 2} \sum_{s,t} z_{it}z_{jt}z_{is}z_{1s} \right|$$

$$\leq \lim_{p\to\infty} \frac{4\sqrt{K_6}n^{1/2}}{p} \left| \max_{j\geq 2} \sum_{i\geq 2} \sum_{s,t} n^{-1}\mathbb{E}[z_i^2 z_j z_1] + \frac{n-1}{n}\mathbb{E}[z_i z_j]\mathbb{E}[z_i z_1] \right|$$

$$= 0.$$

Finally, for term (7), we have

$$\lim_{p\to\infty} \left| \frac{1}{pn^2} \sum_{i\geq 2, j\geq 2} \sum_{s,t} 4\varepsilon_i^2 \varepsilon_j \sqrt{1-\varepsilon_j^2} \cdot y_{1s}^2 y_{1t}y_{jt} \right|$$

$$\leq \lim_{p\to\infty} \frac{4K_6 n^{-k}}{pn^2} \sum_{j\geq 2} |\varepsilon_j| \left| \sum_{s,t} y_{1s}^2 y_{1t}y_{jt} \right|$$

$$\leq \lim_{p\to\infty} \frac{4K_6^{3/2}n^{1/2}}{pn^2} \max_{j\geq 2} \left| \sum_s z_{1s}^3 z_{js} + \sum_{s,t\neq s} z_{1s}^2 z_{1t}z_{jt} \right|$$

$$= \lim_{p \to \infty} \frac{4n^{1/2}}{p} \max_{j \geq 2} \left| n^{-1}\mathbb{E}[y_1^3 y_j] + \frac{n-1}{n}\mathbb{E}[y_1^2]\mathbb{E}[y_1 y_j] \right|$$

$$= 0.$$

Let us continue with the fourth and last term in eq. (12):

$$\lim_{p \to \infty} p^{-1} \sum_{i,j \geq 2} S_{\widehat{oc},ij} T_{\widehat{oc},ij} = \lim_{p \to \infty} p^{-1} \sum_{i \geq 2} S_{\widehat{oc},ii} T_{\widehat{oc},ii}$$

$$= \lim_{p \to \infty} p^{-2} n^{-2} \sum_{i \geq 2} \sum_{t} u_{it}^2 \sum_{k \geq 2} \sum_{s} u_{ks}^2$$

$$= \lim_{p \to \infty} p^{-2} n^{-2} \sum_{i \geq 2} \sum_{t} \left( y_{it}\sqrt{1-\varepsilon_i^2} + y_{1t}\varepsilon_i \right)^2 \sum_{k \geq 2} \sum_{s} \left( y_{ks}\sqrt{1-\varepsilon_k^2} + y_{1s}\varepsilon_k \right)^2$$

$$= \lim_{p \to \infty} p^{-2} n^{-2} \sum_{i \geq 2} \sum_{t} \left( y_{it}^2(1-\varepsilon_i^2) + y_{1t}^2\varepsilon_i^2 + 2y_{it}y_{1t}\sqrt{1-\varepsilon_i^2}\varepsilon_i \right)$$

$$\cdot \sum_{k \geq 2} \sum_{s} \left( y_{ks}^2(1-\varepsilon_k^2) + y_{1s}^2\varepsilon_k^2 + 2y_{ks}y_{1s}\sqrt{1-\varepsilon_k^2}\varepsilon_k \right)$$

$$= \lim_{p \to \infty} p^{-2} n^{-2} \sum_{i,k \geq 2} \sum_{s,t} \left\{ y_{it}^2 y_{ks}^2 \overset{(2)}{+} y_{it}^2 y_{ks}^2 (\varepsilon_i^2\varepsilon_k^2 - \varepsilon_i^2 - \varepsilon_k^2) \right.$$

$$\overset{(3)}{+} y_{1t}^2 y_{1s}^2 \varepsilon_i^2 \varepsilon_k^2$$

$$\overset{(4)}{+} 4y_{it}y_{1t}y_{ks}y_{1s}\sqrt{1-\varepsilon_i^2}\varepsilon_i\sqrt{1-\varepsilon_k^2}\varepsilon_k$$

$$\overset{(5)}{+} 2y_{1t}^2 y_{ks}^2 \varepsilon_i^2(1-\varepsilon_k^2)$$

$$\overset{(6)}{+} 4y_{it}^2 y_{ks}y_{1s}(1-\varepsilon_i^2)\sqrt{1-\varepsilon_k^2}\varepsilon_k$$

$$\overset{(7)}{+} 4y_{1t}^2 y_{ks}y_{1s}\varepsilon_i^2\sqrt{1-\varepsilon_k^2}\varepsilon_k.$$

The first term is equal to $S_{oc,ij}T_{oc,ij}$, we therefore have to show that the remaining terms (2)-(7) go to zero. First note that the terms (3), (4) and (7) already appeared in the third term in eq. (12). For term (2), we have

$$\lim_{p \to \infty} p^{-2} n^{-1} \left| \sum_{i,k \geq 2} \sum_{s,t} y_{it}^2 y_{ks}^2 (\varepsilon_i^2\varepsilon_k^2 - \varepsilon_i^2 - \varepsilon_k^2) \right|$$

$$\leq \lim_{p \to \infty} 2p^{-2} n^{-2} \sum_{i,k \geq 2} \left| \sum_{s,t} y_{it}^2 y_{ks}^2 \varepsilon_i^2 \right|$$

$$\leq \lim_{p \to \infty} 2p^{-2} n^{-2} K_6 n^{-k} \max_{i \geq 2} \left| \sum_{k \geq 2} \sum_{s,t} y_{it}^2 y_{ks}^2 \right|$$

$$\leq \lim_{p \to \infty} 2p^{-2} n^{-2} K_6 n^{-k} \max_{i \geq 2} \left| \sum_{k \geq 2} \sum_{s} y_{is}^2 y_{ks}^2 + \sum_{s,t \neq s} y_{it}^2 y_{ks}^2 \right|$$

$$\leq \lim_{p \to \infty} 2p^{-1} K_6 n^{-k} \frac{1}{p} \sum_{k} \max_{i \geq 2} \left\{ n^{-1}\mathbb{E}[y_i^2 y_k^2] + \frac{n-1}{n}\mathbb{E}[y_i^2]\mathbb{E}[y_k^2] \right\}$$

$$= 0.$$

Term (5) is

$$\lim_{p \to \infty} p^{-2} n^{-2} \left| \sum_{i,k \geq 2} \sum_{s,t} 2y_{1t}^2 y_{ks}^2 \varepsilon_i^2 (1-\varepsilon_k^2) \right|$$

$$\leq \lim_{p\to\infty} 2p^{-2}n^{-2}K_6 n^{-k}\sum_{k\geq 2}\left|\sum_{s,t}y_{1t}^2 y_{ks}^2\right|$$

$$\leq \lim_{p\to\infty} 2p^{-1}K_6\frac{1}{p}\sum_{k\geq 2}\left\{n^{-1}\mathbb{E}[z_1^2 z_k^2]+\frac{n-1}{n}\mathbb{E}[z_1^2]\mathbb{E}[z_k^2]\right\}$$

$$= 0.$$

and, finally, term (6) is

$$\lim_{p\to\infty}p^{-2}n^{-2}\left|\sum_{i,k\geq 2}\sum_{s,t}4y_{it}^2 y_{ks}y_{1s}(1-\varepsilon_i^2)\sqrt{1-\varepsilon_k^2}\,\varepsilon_k\right|$$

$$\leq \lim_{p\to\infty}4p^{-2}n^{-2}\sum_{i,k\geq 2}|\varepsilon_k|\left|\sum_{s,t}y_{it}^2 y_{ks}y_{1s}\right|$$

$$\leq \lim_{p\to\infty}4p^{-1}\sqrt{K_6}n^{1/2}\frac{1}{p}\sum_{i\geq 2}\max_{k\geq 2}\left\{n^{-1}\sum_{i\geq 2}\mathbb{E}[z_i^2 z_k z_1]+\frac{n-1}{n}\mathbb{E}[z_i^2]\mathbb{E}[z_k z_1]\right\}$$

$$= 0.$$

As we have shown that all terms eq. (12) converge to the corresponding terms in the true orthogonal complement, the same holds for $W_{\widehat{oc}}$. $\qquad\square$

**Lemma 7.** *Under the above assumptions,*

$$\lim_{p\to\infty}W_{oc}=\lim_{p\to\infty}V_{oc}.$$

*Proof of Lemma 7.*

$$\lim_{p\to\infty}W_{oc}=\lim_{p\to\infty}\sum_{i,j}\frac{1}{p}(C_{oc,ij}-S_{oc,ij})(S_{oc,ij}-T_{oc,ij})$$

$$=\lim_{p\to\infty}p^{-1}\sum_{i,j}C_{oc,ij}S_{oc,ij}-C_{oc,ij}T_{oc,ij}-S_{oc,ij}^2+S_{oc,ij}T_{oc,ij}$$

$$=\lim_{p\to\infty}p^{-1}\sum_{i,j}S_{oc,ij}T_{oc,ij}-C_{oc,ij}T_{oc,ij}-(S_{oc,ij}^2-C_{oc,ij}S_{oc,ij})$$

$$\overset{(a)}{=}\lim_{p\to\infty}p^{-1}\sum_{i,j}\mathbb{E}[S_{oc,ij}T_{oc,ij}]-\mathbb{E}[S_{oc,ij}]\mathbb{E}[T_{oc,ij}]-(\mathbb{E}[S_{oc,ij}^2]-\mathbb{E}^2[S_{oc,ij}])$$

$$=\lim_{p\to\infty}p^{-1}\sum_{i,j}\mathrm{Cov}(S_{oc,ij},T_{oc,ij})-\mathrm{Var}(S_{oc,ij})$$

$$=\lim_{p\to\infty}V_{oc}.$$

As we see, for the lemma to hold we have to show that equality (a) holds. Convergence of the first and third term to their expectations was already necessary for the consistency of analytic shrinkage. It remains to be shown that the variance of the second and the fourth term goes to zero.

The variance of the second term in the limit is

$$\lim_{p\to\infty}\mathrm{Var}\left(p^{-1}\sum_{i,j}C_{oc,ij}T_{oc,ij}\right)=\lim_{p\to\infty}\frac{1}{p^2}\mathrm{Var}\left(\sum_i C_{oc,ii}T_{oc,ii}\right)$$

$$=\lim_{p\to\infty}\frac{1}{p^2}\sum_{i,j}C_{oc,ii}C_{oc,jj}\mathrm{Cov}\left(T_{oc,ii},T_{oc,jj}\right)$$

$$=\lim_{p\to\infty}\frac{1}{p^2}\sum_{i,j}C_{oc,ii}C_{oc,jj}\mathrm{Cov}\left(\frac{1}{np}\sum_{ks}y_{ks}^2,\frac{1}{np}\sum_{lt}y_{lt}^2\right)$$

$$= \lim_{p \to \infty} \frac{1}{p^4 n} \sum_{i,j,k,l} C_{oc,ii} C_{oc,jj} \mathrm{Cov}\left(y_k^2, y_l^2\right)$$

$$= 0,$$

and the variance of the fourth term in the limit is

$$\lim_{p \to \infty} \mathrm{Var}\left(p^{-1} \sum_{i,j} C_{oc,ij} S_{oc,ij}\right) = \lim_{p \to \infty} \frac{1}{p^2} \mathrm{Var}\left(\sum_i C_{oc,ii} S_{oc,ii}\right)$$

$$= \lim_{p \to \infty} \frac{1}{p^2} \sum_{i,j} C_{oc,ii} C_{oc,jj} \mathrm{Cov}\left(S_{oc,ii}, S_{oc,jj}\right)$$

$$= \lim_{p \to \infty} \frac{1}{p^2} \sum_{i,j} C_{oc,ii} C_{oc,jj} \mathrm{Cov}\left(\frac{1}{n}\sum_s y_{is}^2, \frac{1}{n}\sum_{jt} y_{jt}^2\right)$$

$$= \lim_{p \to \infty} \frac{1}{p^2 n} \sum_{i,j} C_{oc,ii} C_{oc,jj} \mathrm{Cov}\left(y_i^2, y_j^2\right)$$

$$= 0.$$

$\square$

**Lemma 8.** *Under the above assumptions,*

$$\lim_{p \to \infty} \hat{V}_{oc} = \lim_{p \to \infty} \hat{V}_{\widehat{oc}}.$$

*Proof of Lemma 8.* Let us now look at the sum over the $\widehat{\mathrm{Var}}(S_{ij}'')$, the estimated variance of the covariance matrix in the basis defined above:

$$\frac{1}{p} \sum_{i \geq 2, j \geq 2} \widehat{\mathrm{Var}}(S_{ij}'') = \frac{1}{p(n-1)n} \sum_{i \geq 2, j \geq 2} \sum_s \left(u_{is} u_{js} - \frac{1}{n} \sum_{t'} u_{it'} u_{jt'}\right)^2$$

$$= \frac{1}{p(n-1)n} \sum_{i \geq 2, j \geq 2} \left(\sum_s u_{is}^2 u_{js}^2 - \frac{1}{n} \sum_{s,t} u_{it} u_{jt} u_{is} u_{js}\right).$$

Note that the second term was already analysed in the third term of eq. (12). We therefore only have to analyse the first term:

$$\frac{1}{p(n-1)n} \sum_{i \geq 2, j \geq 2} \sum_t u_{it}^2 u_{jt}^2 = \frac{1}{p(n-1)n} \sum_{i \geq 2, j \geq 2} \sum_t \left(y_{it}\sqrt{1-\varepsilon_i^2} + y_{1t}\varepsilon_i\right)^2 \left(y_{jt}\sqrt{1-\varepsilon_j^2} + y_{1t}\varepsilon_j\right)^2$$

$$= \frac{1}{p(n-1)n} \sum_{i \geq 2, j \geq 2} \sum_t \left(y_{it}^2(1-\varepsilon_i^2) + y_{it}y_{1t}\varepsilon_i\sqrt{1-\varepsilon_i^2} + y_{1t}^2\varepsilon_i^2\right)$$

$$\cdot \left(y_{jt}^2(1-\varepsilon_j^2) + y_{jt}y_{1t}\varepsilon_j\sqrt{1-\varepsilon_j^2} + y_{1t}^2\varepsilon_j^2\right)$$

$$= \frac{1}{p(n-1)n} \sum_{i \geq 2, j \geq 2} \sum_t \Bigg\{ y_{it}^2 y_{jt}^2 \overset{(2)}{+} y_{it}^2 y_{jt}^2 (\varepsilon_i^2 \varepsilon_j^2 - \varepsilon_i^2 - \varepsilon_j^2)$$

$$\overset{(3)}{+} y_{1t}^4 \varepsilon_i^2 \varepsilon_j^2$$

$$\overset{(4)}{+} 4 y_{it} y_{jt} y_{1t}^2 \sqrt{1-\varepsilon_i^2}\varepsilon_i \sqrt{1-\varepsilon_j^2}\varepsilon_j$$

$$\overset{(5)}{+} 2 y_{1t}^2 y_{jt}^2 \varepsilon_i^2 (1-\varepsilon_j^2)$$

$$\overset{(6)}{+} 4 y_{it}^2 y_{jt} y_{1t} (1-\varepsilon_i^2)\sqrt{1-\varepsilon_j^2}\varepsilon_j$$

$$\overset{(7)}{+} 4 y_{1t}^3 y_{jt} \varepsilon_i^2 \sqrt{1-\varepsilon_j^2}\varepsilon_j.$$

For term (2), we have

$$\lim_{p\to\infty}\left|\frac{1}{pn(n-1)}\sum_{i\geq2,j\geq2}\sum_t y_{it}^2 y_{jt}^2(\varepsilon_i^2\varepsilon_j^2-\varepsilon_i^2-\varepsilon_j^2)\right|$$

$$\leq\lim_{p\to\infty}\frac{1}{pn(n-1)}\sum_{i\geq2,j\geq2}\left|\sum_t y_{it}^2 y_{jt}^2(\varepsilon_i^2\varepsilon_j^2-\varepsilon_i^2-\varepsilon_j^2)\right|$$

$$\leq\lim_{p\to\infty}\frac{2K_6 n^{-k}}{n(n-1)}\max_{i\geq2,j\leq2}\left(\sum_t y_{it}^4+y_{jt}^4\right)$$

$$\leq\lim_{p\to\infty}\frac{4K_6 n^{-k}}{n(n-1)}\max_{i\geq2}\left(\sum_t y_{it}^4\right)$$

$$=\lim_{p\to\infty}\frac{4K_6 n^{-k}}{n-1}\max_{i\geq2}\mathbb{E}[y_i^4]$$

$$=0.$$

Term (3) is

$$\lim_{p\to\infty}\left|\frac{1}{pn(n-1)}\sum_{i\geq2,j\geq2}\sum_t y_{1t}^4\varepsilon_i^2\varepsilon_j^2\right|$$

$$\leq\lim_{p\to\infty}\frac{K_6^2 n^{-2k}}{pn(n-1)}\left|\sum_t y_{1t}^4\right|$$

$$=\lim_{p\to\infty}\frac{K_6^2 n^{-2k}}{p(n-1)}\mathbb{E}[y_1^4]$$

$$=0.$$

Term (4) is

$$\lim_{p\to\infty}\left|\frac{1}{pn(n-1)}\sum_{i\geq2,j\geq2}\sum_t 4y_{it}y_{jt}y_{1t}^2\sqrt{1-\varepsilon_i^2}\varepsilon_i\sqrt{1-\varepsilon_j^2}\varepsilon_j\right|$$

$$\leq\lim_{p\to\infty}\frac{4}{pn(n-1)}\sum_{i\geq2,j\geq2}|\varepsilon_i\varepsilon_j|\left|\sum_t y_{it}y_{jt}y_{1t}^2\right|$$

$$\leq\lim_{p\to\infty}\frac{4}{pn(n-1)}\left\{\sum_{i\geq2}\varepsilon_i^2\left|\sum_t y_{it}^2 y_{1t}^2\right|+2\sum_{i\geq2,j\geq2\neq i}\varepsilon_i^2\left|\sum_t y_{it}y_{jt}y_{1t}^2\right|\right\}$$

$$\leq\lim_{p\to\infty}\frac{4K_6 n^{-k}}{pn(n-1)}\left\{\max_{i\geq2}\left|\sum_t y_{it}^2 y_{1t}^2\right|+\max_{i\geq2}\left|\sum_{j\geq2\neq i}\sum_t y_{it}y_{jt}y_{1t}^2\right|\right\}$$

$$\leq\lim_{p\to\infty}\frac{4K_6 n^{-k}}{p(n-1)}\left\{\max_{i\geq2}\mathbb{E}[y_i^2 y_1^2]+\max_{i\geq2}\sum_{j\geq2\neq i}\left|\mathbb{E}[y_i y_j y_1^2]\right|\right\}$$

$$=0.$$

For term (5), we have

$$\lim_{p\to\infty}\left|\frac{1}{pn(n-1)}\sum_{i\geq2,j\geq2}\sum_t 2y_{1t}^2 y_{jt}^2\varepsilon_i^2(1-\varepsilon_j^2)\right|$$

$$\leq\lim_{p\to\infty}\frac{2K_6 n^{-k}}{pn(n-1)}\sum_{j\geq2}\left|\sum_t y_{1t}^2 y_{jt}^2\right|$$

$$\leq \lim_{p\to\infty} \frac{2K_6 n^{-k}}{p(n-1)} \sum_{j\geq 2} \mathbb{E}[y_1^2 y_j^2]$$

$$= 0.$$

and for term (6)

$$\lim_{p\to\infty} \left| \frac{1}{pn(n-1)} \sum_{i\geq 2, j\geq 2} \sum_t 4y_{it}^2 y_{jt} y_{1t}(1-\varepsilon_i^2)\sqrt{1-\varepsilon_j^2}\varepsilon_j \right|$$

$$\leq \lim_{p\to\infty} \frac{4}{pn(n-1)} \sum_{i\geq 2, j\geq 2} |\varepsilon_j| \left| \sum_t y_{it}^2 y_{jt} y_{1t} \right|$$

$$\leq \lim_{p\to\infty} \frac{4\sqrt{K_6}n^{1/2}}{pn(n-1)} \sum_{i\geq 2} \max_{j\geq 2} \left| \sum_t z_{it}^2 z_{jt} z_{1t} \right|$$

$$\leq \lim_{p\to\infty} \frac{4\sqrt{K_6}n^{1/2}}{p(n-1)} \sum_{i\geq 2} \max_{j\geq 2} \mathbb{E}[z_i^2 z_j z_1]$$

$$= 0.$$

Finally, for term (7), we have

$$\lim_{p\to\infty} \left| \frac{1}{pn(n-1)} \sum_{i\geq 2, j\geq 2} \sum_t 4y_{1t}^3 y_{jt}\varepsilon_i^2 \sqrt{1-\varepsilon_j^2}\varepsilon_j \right|$$

$$\leq \lim_{p\to\infty} \frac{4}{pn(n-1)} \sum_{i\geq 2, j\geq 2} \varepsilon_i^2 |\varepsilon_j| \left| \sum_t y_{1t}^3 y_{jt} \right|$$

$$\leq \lim_{p\to\infty} \frac{K_6^{3/2}n^{1/2}}{pn(n-1)} \max_{j\geq 2} \left| \sum_t z_{it}^3 z_{jt} \right|$$

$$\leq \lim_{p\to\infty} \frac{\sqrt{K_6}n^{1/2}}{p(n-1)} \max_{j\geq 2} \mathbb{E}[z_1^3 z_j]$$

$$= 0.$$

With this, it is shown that the estimate of $V$ converges to the same value for the estimated and the true orthogal complement. $\qquad\square$

## 2 Algorithm Listing

---

**Input:** data $X$, eigenvalues $l$, eigenvectors $V$
**Output:** aoc-shrinkage covariance matrix $C^{aoc-shr}$
  1: $r^* = 0, r = 1$
  2: $\lambda_{old} = \text{calc\_lambda}(X)$
  3: $\nu_{old} = \text{mean}(l)$
  4: **while** TRUE
  5:       $X_{oc} = V(:, r+1:p)^{\text{T}} X$
  6:       $l_{oc} = l(r+1:p)$
  7:       $\lambda_{new} = \text{calc\_lambda}(X_{oc})$
  8:       $\nu_{new} = l(:, r+1:p)$
  9:       $\Delta_R = \text{calc\_Delta\_R}(X_{oc}, \lambda_{new}, \nu_{new}, \lambda_{old}, \nu_{old})$
10:       $\sigma_R = \text{calc\_sigma\_R}(l_{oc}, \lambda_{new}, \nu_{new}, \lambda_{old}, \nu_{old})$
11:       $\sigma_E = \text{calc\_sigma\_E}(l_{oc})$
12:       **if** $\Delta_R - m_\Delta \sigma_R - m_{E_{oc}} \lambda_{new}^2 \sigma_E > 0$
13:           $r^* = r^* + 1$
14:           $r = r + 1$
15:           $\lambda_{old} = \lambda_{new}$
16:           $\nu_{old} = \nu_{new}$
17:       **else**
18:           BREAK
19:       **end**
20: **end**
21: $P_{pca} = V(:, 1:r^*)^{\text{T}} V(:, 1:r^*)$
22: $P_{oc} = V(:, r^*+1:p)^{\text{T}} V(:, r^*+1:p)$
23: $C^{sh} = (1 - \lambda_{old}) \cdot S + \lambda_{old} \nu_{old} \text{eye}(p)$
24: **return** $C^{aoc-sh} = P_{pca}^{\text{T}} S P_{pca} + P_{oc}^{\text{T}} C^{sh} P_{oc}$

---

## 3 Analytic formulas to conservatively estimate the oc-shrinkage improvement

In order to evaluate the conservative estimate of the performance difference between standard shrinkage and *oc*-shrinkage, we need variance estimates for numerator

$$\hat{V} := \frac{1}{p} \sum_{i,j} \left\{ \widehat{\text{Var}}\left(S_{ij}\right) - \widehat{\text{Cov}}\left(S_{ij}, T_{ij}\right) \right\}$$

and denominator

$$\hat{E} := \frac{1}{p} \sum_{i,j} (S_{ij} - T_{ij})^2$$

in the analytic shrinkage formula, $\hat{\sigma}_{\hat{V}}^2$ and $\hat{\sigma}_{\hat{E}}^2$, and an estimate for correlation between these terms,

$$\text{corr}\left(\hat{V}_{\widehat{oc}}, \hat{E}_{\widehat{oc}}\right) = \text{Cov}\left(\hat{V}_{\widehat{oc}}, \hat{E}_{\widehat{oc}}\right) / \hat{\sigma}_{\hat{V}} \hat{\sigma}_{\hat{E}}.$$

We first derive expressions only using the i.i.d. assumption, then simplify by assuming normality and considering all expressions in the eigenbasis. The resulting formulas are quite long, but fast to evaluate numerically.

### 3.1 Variance of $\hat{V} = \frac{1}{p} \sum_{i,j} \left\{ \widehat{\text{Var}}\left(S_{ij}\right) - \widehat{\text{Cov}}\left(S_{ij}, T_{ij}\right) \right\}$

The variance of $\hat{V}$ is given by

$$\text{Var}(\hat{V}) = \text{Var}\left(\frac{1}{p} \sum_{i,j} \left\{ \widehat{\text{Var}}\left(S_{ij}\right) - \widehat{\text{Cov}}\left(S_{ij}, T_{ij}\right) \right\}\right)$$

$$= \frac{1}{p^2}\mathrm{Var}\left(\frac{1}{(n-1)n^2}\sum_{i,j}\left\{\sum_t x_{it}^2 x_{jt}^2 - \frac{1}{n}\sum_{s,t} x_{it}x_{jt}x_{is}x_{js}\right\}\right.$$

$$\left. - \frac{1}{p(n-1)n}\sum_{i,j}\left\{\sum_t x_{it}^2 x_{jt}^2 - \frac{1}{n}\sum_{s,t} x_{is}^2 x_{jt}^2\right\}\right)$$

$$= \frac{1}{p^2(n-1)^2 n^2}\mathrm{Var}\left(\sum_{i,j}\left\{\frac{p-1}{p}\sum_t x_{it}^2 x_{jt}^2 - \frac{1}{n}\sum_{s,t} x_{it}x_{jt}x_{is}x_{js} + \frac{1}{np}\sum_{s,t} x_{is}^2 x_{jt}^2\right\}\right)$$

$$= \frac{1}{p^2(n-1)^2 n^2}\sum_{i,j,k,l}\left\{\frac{n(p-1)}{p}\mathrm{Cov}(x_{i1}^2 x_{j1}^2, x_{k1}^2 x_{l1}^2)\right.$$

$$+ \frac{1}{n^2}\mathrm{Cov}\left(\sum_{t,s} x_{it}x_{jt}x_{is}x_{js}, \sum_{t',s'} x_{kt'}x_{lt'}x_{ks'}x_{ls'}\right)$$

$$+ \frac{1}{n^2 p^2}\mathrm{Cov}\left(\sum_{t,s} x_{it}^2 x_{js}^2, \sum_{t',s'} x_{kt'}^2 x_{ls'}^2\right) - \frac{2(p-1)}{np}\mathrm{Cov}\left(\sum_t x_{it}^2 x_{jt}^2, \sum_{t',s} x_{kt'}x_{lt'}x_{ks}x_{ls}\right)$$

$$\left. + \frac{2(p-1)}{np^2}\mathrm{Cov}\left(\sum_t x_{it}^2 x_{jt}^2, \sum_{t',s'} x_{kt'}^2 x_{ls'}^2\right) - \frac{2}{n^2 p}\mathrm{Cov}\left(\sum_{t,s} x_{it}x_{jt}x_{is}x_{js}, \sum_{t',s'} x_{kt'}^2 x_{ls'}^2\right)\right\}.$$

Under the i.i.d. assumption, this expression simplifies:

$$= \frac{1}{p^2(n-1)^2 n^2}\sum_{i,j,k,l}\left\{\frac{n(p-1)}{p}\mathrm{Cov}(x_{i1}^2 x_{j1}^2, x_{k1}^2 x_{l1}^2)\right.$$

$$+ \frac{1}{n^2}\left\{n\mathrm{Cov}(x_{i1}^2 x_{j1}^2, x_{k1}^2 x_{l1}^2) + 4n(n-1)\mathrm{Cov}\left(x_{i1}^2 x_{j1}^2, x_{k1}x_{l1}x_{k2}x_{l2}\right)\right.$$

$$+ 2n(n-1)\,\mathrm{Cov}\left(x_{i1}x_{j1}x_{i2}x_{j2}, x_{k1}x_{l1}x_{k2}x_{l2}\right)$$

$$\left. + 4n(n-1)(n-2)\mathrm{Cov}\left(x_{i1}x_{j1}x_{i2}x_{j2}, x_{k1}x_{l1}x_{k3}x_{l3}\right)\right\}$$

$$+ \frac{1}{n^2 p^2}\left\{n\mathrm{Cov}(x_{i1}^2 x_{j1}^2, x_{k1}^2 x_{l1}^2) + 4n(n-1)\mathrm{Cov}\left(x_{i1}^2 x_{j1}^2, x_{k1}^2 x_{l2}^2\right)\right.$$

$$+ 2n(n-1)\,\mathrm{Cov}\left(x_{i1}^2 x_{j2}^2, x_{k1}^2 x_{l2}^2\right)$$

$$\left. + 4n(n-1)(n-2)\mathrm{Cov}\left(x_{i1}^2 x_{j2}^2, x_{k1}^2 x_{l3}^2\right)\right\}$$

$$- \frac{2(p-1)}{np}\left\{n\cdot\mathrm{Cov}\left(x_{i1}^2 x_{j1}^2, x_{k1}^2 x_{l1}^2\right) + 2n(n-1)\mathrm{Cov}\left(x_{i1}^2 x_{j1}^2, x_{k1}x_{l1}x_{k2}x_{l2}\right)\right\}$$

$$- \frac{2(p-1)}{np^2}\left\{n\cdot\mathrm{Cov}\left(x_{i1}^2 x_{j1}^2, x_{k1}^2 x_{l1}^2\right) + 2n(n-1)\mathrm{Cov}\left(x_{i1}^2 x_{j1}^2, x_{k1}^2 x_{l2}^2\right)\right\}$$

$$+ \frac{2}{n^2 p}\left\{n\mathrm{Cov}(x_{i1}^2 x_{j1}^2, x_{k1}^2 x_{l1}^2) + 2n(n-1)\mathrm{Cov}\left(x_{i1}^2 x_{j1}^2, x_{k1}x_{l1}x_{k2}x_{l2}\right)\right.$$

$$+ 2n(n-1)\mathrm{Cov}\left(x_{i1}^2 x_{j1}^2, x_{k1}^2 x_{l2}^2\right) + 2n(n-1)\,\mathrm{Cov}\left(x_{i1}x_{j1}x_{i2}x_{j2}, x_{k1}^2 x_{l2}^2\right)$$

$$\left. + 4n(n-1)(n-2)\mathrm{Cov}\left(x_{i1}x_{j1}x_{i2}x_{j2}, x_{k1}^2 x_{l3}^2\right)\right\}$$

$$= \frac{1}{p^2(n-1)^2 n^2}\sum_{i,j,k,l}\left\{\frac{n^2 p^2 - n^2 p - 2np^2 + np + 2n + 2p + 1}{np^2}\mathrm{Cov}(x_{i1}^2 x_{j1}^2, x_{k1}^2 x_{l1}^2)\right.$$

$$- \frac{4(n-1)(np - n - p - 1)}{np}\mathrm{Cov}\left(x_{i1}^2 x_{j1}^2, x_{k1}x_{l1}x_{k2}x_{l2}\right)$$

$$+ \frac{2(n-1)}{n}\,\mathrm{Cov}\left(x_{i1}x_{j1}x_{i2}x_{j2}, x_{k1}x_{l1}x_{k2}x_{l2}\right)$$

$$+ \frac{4(n-1)(n-2)}{n}\mathrm{Cov}\left(x_{i1}x_{j1}x_{i2}x_{j2}, x_{k1}x_{l1}x_{k3}x_{l3}\right)\Big\}$$

$$- \frac{4(n-1)(np-n-p-1)}{np^2}\mathrm{Cov}\left(x_{i1}^2x_{j1}^2, x_{k1}^2x_{l2}^2\right)$$

$$+ \frac{2(n-1)}{np^2}\mathrm{Cov}\left(x_{i1}^2x_{j2}^2, x_{k1}^2x_{l2}^2\right)$$

$$+ \frac{4(n-1)(n-2)}{np^2}\mathrm{Cov}\left(x_{i1}^2x_{j2}^2, x_{k1}^2x_{l3}^2\right)$$

$$+ \frac{4(n-1)}{np}\mathrm{Cov}\left(x_{i1}x_{j1}x_{i2}x_{j2}, x_{k1}^2x_{l2}^2\right)$$

$$+ \frac{8(n-1)(n-2)}{np}\mathrm{Cov}\left(x_{i1}x_{j1}x_{i2}x_{j2}, x_{k1}^2x_{l3}^2\right).$$

For the nine different covariance expressions, Appendix 3.4 contains evaluations under normality.

### 3.2 Variance of $\hat{E} = \frac{1}{p}\sum_{i,j}(S_{ij} - T_{ij})^2$

For $\hat{E}$, we have to calculate

$$\mathrm{Var}\left(\hat{E}\right) = \mathrm{Var}\left(\frac{1}{p}\sum_{i,j}(S_{ij} - T_{ij})^2\right)$$

$$= \mathrm{Var}\left(\frac{1}{pn^2}\sum_{i,j}\left\{\sum_{s,t}x_{is}x_{js}x_{it}x_{jt} - \frac{2}{p}\delta_{ij}\sum_{k}\sum_{s,t}x_{is}x_{js}x_{kt}x_{kt} + \frac{1}{p^2}\delta_{ij}\sum_{k,l}\sum_{s,t}x_{ks}x_{ks}x_{lt}x_{lt}\right\}\right)$$

$$= \mathrm{Var}\left(\frac{1}{pn^2}\sum_{i,j}\left\{\sum_{s,t}x_{is}x_{js}x_{it}x_{jt} - \frac{1}{p}\sum_{s,t}x_{is}^2x_{jt}^2\right\}\right)$$

$$= \frac{1}{p^2n^4}\sum_{i,j,k,l}\left\{\mathrm{Cov}\left(\sum_{s,t}x_{is}x_{js}x_{it}x_{jt}, \sum_{s,t}x_{is}x_{js}x_{it}x_{jt}\right) - \frac{2}{p}\mathrm{Cov}\left(\sum_{s,t}x_{is}x_{js}x_{it}x_{jt}, \sum_{s,t}x_{is}^2x_{jt}^2\right)\right.$$

$$\left. + \frac{1}{p^2}\mathrm{Cov}\left(\sum_{s,t}x_{is}^2x_{jt}^2, \sum_{s,t}x_{is}^2x_{jt}^2\right)\right\}$$

Again, this expression simplifies under the i.i.d. assumption:

$$= \frac{1}{p^2n^4}\sum_{i,j,k,l}\left\{n\mathrm{Cov}(x_{i1}^2x_{j1}^2, x_{k1}^2x_{l1}^2) + 4n(n-1)\mathrm{Cov}\left(x_{i1}^2x_{j1}^2, x_{k1}x_{l1}x_{k2}x_{l2}\right)\right.$$

$$\qquad + 2n(n-1)\ \mathrm{Cov}\left(x_{i1}x_{j1}x_{i2}x_{j2}, x_{k1}x_{l1}x_{k2}x_{l2}\right)$$

$$\qquad + 4n(n-1)(n-2)\mathrm{Cov}\left(x_{i1}x_{j1}x_{i2}x_{j2}, x_{k1}x_{l1}x_{k3}x_{l3}\right)$$

$$\qquad - \frac{2}{p}\Big\{n\mathrm{Cov}(x_{i1}^2x_{j1}^2, x_{k1}^2x_{l1}^2) + 2n(n-1)\mathrm{Cov}\left(x_{i1}^2x_{j1}^2, x_{k1}x_{l1}x_{k2}x_{l2}\right)$$

$$\qquad + 2n(n-1)\mathrm{Cov}\left(x_{i1}^2x_{j1}^2, x_{k1}^2x_{l2}^2\right) + 2n(n-1)\ \mathrm{Cov}\left(x_{i1}x_{j1}x_{i2}x_{j2}, x_{k1}^2x_{l2}^2\right)$$

$$\qquad + 4n(n-1)(n-2)\mathrm{Cov}\left(x_{i1}x_{j1}x_{i2}x_{j2}, x_{k1}^2x_{l3}^2\right)\Big\}$$

$$\qquad + \frac{1}{p^2}\Big\{n\mathrm{Cov}(x_{i1}^2x_{j1}^2, x_{k1}^2x_{l1}^2) + 4n(n-1)\mathrm{Cov}\left(x_{i1}^2x_{j1}^2, x_{k1}^2x_{l2}^2\right)$$

$$\qquad + 2n(n-1)\ \mathrm{Cov}\left(x_{i1}^2x_{j2}^2, x_{k1}^2x_{l2}^2\right)$$

$$\qquad\left. + 4n(n-1)(n-2)\mathrm{Cov}\left(x_{i1}^2x_{j2}^2, x_{k1}^2x_{l3}^2\right)\Big\}\right\}.$$

Collecting terms, we obtain

$$
= \frac{1}{p^2 n^4} \sum_{i,j,k,l} \left\{ \frac{np^2 - 2pn + n}{p^2} \mathrm{Cov}(x_{i1}^2 x_{j1}^2, x_{k1}^2 x_{l1}^2) \right.
$$

$$
- \frac{4n(n-1)(p-1)}{p} \mathrm{Cov}\left( x_{i1}^2 x_{j1}^2, x_{k1} x_{l1} x_{k2} x_{l2} \right)
$$

$$
+ 2n(n-1)\, \mathrm{Cov}\left( x_{i1} x_{j1} x_{i2} x_{j2}, x_{k1} x_{l1} x_{k2} x_{l2} \right)
$$

$$
\left. + 4n(n-1)(n-2)\mathrm{Cov}\left( x_{i1} x_{j1} x_{i2} x_{j2}, x_{k1} x_{l1} x_{k3} x_{l3} \right) \right\}
$$

$$
- \frac{4n(n-1)(p-1)}{p^2} \mathrm{Cov}\left( x_{i1}^2 x_{j1}^2, x_{k1}^2 x_{l2}^2 \right)
$$

$$
+ \frac{2n(n-1)}{p^2} \mathrm{Cov}\left( x_{i1}^2 x_{j2}^2, x_{k1}^2 x_{l2}^2 \right)
$$

$$
+ \frac{4n(n-1)(n-2)}{p^2} \mathrm{Cov}\left( x_{i1}^2 x_{j2}^2, x_{k1}^2 x_{l3}^2 \right)
$$

$$
- \frac{4n(n-1)}{p} \mathrm{Cov}\left( x_{i1} x_{j1} x_{i2} x_{j2}, x_{k1}^2 x_{l2}^2 \right)
$$

$$
- \frac{8n(n-1)(n-2)}{p} \mathrm{Cov}\left( x_{i1} x_{j1} x_{i2} x_{j2}, x_{k1}^2 x_{l3}^2 \right).
$$

The same nine covariance expressions show up. Now, only the calculation of the covariance remains.

### 3.3 Covariance between $\hat{V}$ and $\hat{E}$

For the covariance, we obtain

$$
\mathrm{Cov}(\hat{V}, \hat{E}) = \mathrm{Cov}\left( \frac{1}{p(n-1)n} \sum_{i,j} \left\{ \frac{p-1}{p} \sum_{t} x_{it}^2 x_{jt}^2 - \frac{1}{n} \sum_{s,t} x_{it} x_{jt} x_{is} x_{js} + \frac{1}{np} \sum_{s,t} x_{is}^2 x_{jt}^2 \right\}, \right.
$$

$$
\left. \frac{1}{pn^2} \sum_{i,j} \left\{ \sum_{s,t} x_{is} x_{js} x_{it} x_{jt} - \frac{1}{p} \sum_{s,t} x_{is}^2 x_{jt}^2 \right\} \right)
$$

$$
= \frac{1}{p^2 (n-1) n^3} \sum_{i,j,k,l} \left\{ \frac{(p-1)}{p} \mathrm{Cov}\left( \sum_{t} x_{it}^2 x_{jt}^2, \sum_{t',s} x_{kt'} x_{lt'} x_{ks} x_{ls} \right) \right.
$$

$$
- \frac{(p-1)}{p^2} \mathrm{Cov}\left( \sum_{t} x_{it}^2 x_{jt}^2, \sum_{t',s'} x_{kt'}^2 x_{ls'}^2 \right) - \frac{1}{n} \mathrm{Cov}\left( \sum_{t,s} x_{it} x_{jt} x_{is} x_{js}, \sum_{t',s'} x_{kt'} x_{lt'} x_{ks'} x_{ls'} \right)
$$

$$
\left. + \frac{2}{np} \mathrm{Cov}\left( \sum_{t,s} x_{it} x_{jt} x_{is} x_{js}, \sum_{t',s'} x_{kt'}^2 x_{ls'}^2 \right) - \frac{1}{np^2} \mathrm{Cov}\left( \sum_{t,s} x_{it}^2 x_{js}^2, \sum_{t',s'} x_{kt'}^2 x_{ls'}^2 \right) \right\}.
$$

As before, this simplifies under the i.i.d. assumption:

$$
= \frac{1}{p^2 (n-1) n^3} \sum_{i,j,k,l} \left\{ \frac{(p-1)}{p} \left\{ n \cdot \mathrm{Cov}\left( x_{i1}^2 x_{j1}^2, x_{k1}^2 x_{l1}^2 \right) + 2n(n-1) \mathrm{Cov}\left( x_{i1}^2 x_{j1}^2, x_{k1} x_{l1} x_{k2} x_{l2} \right) \right\} \right.
$$

$$
- \frac{(p-1)}{p^2} \left\{ n \cdot \mathrm{Cov}\left( x_{i1}^2 x_{j1}^2, x_{k1}^2 x_{l1}^2 \right) + 2n(n-1) \mathrm{Cov}\left( x_{i1}^2 x_{j1}^2, x_{k1}^2 x_{l2}^2 \right) \right\}
$$

$$
- \frac{1}{n} \left\{ n \mathrm{Cov}(x_{i1}^2 x_{j1}^2, x_{k1}^2 x_{l1}^2) + 4n(n-1) \mathrm{Cov}\left( x_{i1}^2 x_{j1}^2, x_{k1} x_{l1} x_{k2} x_{l2} \right) \right.
$$

$$
+ 2n(n-1)\, \mathrm{Cov}\left( x_{i1} x_{j1} x_{i2} x_{j2}, x_{k1} x_{l1} x_{k2} x_{l2} \right)
$$

$$
\left. + 4n(n-1)(n-2)\mathrm{Cov}\left( x_{i1} x_{j1} x_{i2} x_{j2}, x_{k1} x_{l1} x_{k3} x_{l3} \right) \right\}
$$

$$+ \frac{2}{np}\Big\{ n\text{Cov}(x_{i1}^2 x_{j1}^2, x_{k1}^2 x_{l1}^2) + 2n(n-1)\text{Cov}\left(x_{i1}^2 x_{j1}^2, x_{k1}x_{l1}x_{k2}x_{l2}\right)$$

$$+ 2n(n-1)\text{Cov}\left(x_{i1}^2 x_{j1}^2, x_{k1}^2 x_{l2}^2\right) + 2n(n-1)\ \text{Cov}\left(x_{i1}x_{j1}x_{i2}x_{j2}, x_{k1}^2 x_{l2}^2\right)$$

$$+ 4n(n-1)(n-2)\text{Cov}\left(x_{i1}x_{j1}x_{i2}x_{j2}, x_{k1}^2 x_{l3}^2\right)\Big\}$$

$$- \frac{1}{np^2}\Big\{ n\text{Cov}(x_{i1}^2 x_{j1}^2, x_{k1}^2 x_{l1}^2) + 4n(n-1)\text{Cov}\left(x_{i1}^2 x_{j1}^2, x_{k1}^2 x_{l2}^2\right)$$

$$+ 2n(n-1)\ \text{Cov}\left(x_{i1}^2 x_{j2}^2, x_{k1}^2 x_{l2}^2\right)$$

$$+ 4n(n-1)(n-2)\text{Cov}\left(x_{i1}^2 x_{j2}^2, x_{k1}^2 x_{l3}^2\right)\Big\}.$$

Collecting terms, we obtain

$$= \frac{1}{p^2(n-1)n^3} \sum_{i,j,k,l}\Bigg\{ \frac{np^2 - 2np - p^2 + n + 2p - 1}{p^2}\text{Cov}(x_{i1}^2 x_{j1}^2, x_{k1}^2 x_{l1}^2)$$

$$+ \frac{2(n-1)(np - n - 2p + 2)}{p}\text{Cov}\left(x_{i1}^2 x_{j1}^2, x_{k1}x_{l1}x_{k2}x_{l2}\right)$$

$$- 2(n-1)\text{Cov}\left(x_{i1}x_{j1}x_{i2}x_{j2}, x_{k1}x_{l1}x_{k2}x_{l2}\right)$$

$$- 4(n-1)(n-2)\text{Cov}\left(x_{i1}x_{j1}x_{i2}x_{j2}, x_{k1}x_{l1}x_{k3}x_{l3}\right)$$

$$- \frac{2(n-1)(np - n - 2p + 1)}{p^2}\text{Cov}\left(x_{i1}^2 x_{j1}^2, x_{k1}^2 x_{l2}^2\right)$$

$$- \frac{2(n-1)}{p^2}\text{Cov}\left(x_{i1}^2 x_{j2}^2, x_{k1}^2 x_{l2}^2\right)$$

$$- \frac{4(n-1)(n-2)}{p^2}\text{Cov}\left(x_{i1}^2 x_{j2}^2, x_{k1}^2 x_{l3}^2\right)$$

$$+ \frac{4(n-1)}{p}\text{Cov}\left(x_{i1}x_{j1}x_{i2}x_{j2}, x_{k1}^2 x_{l2}^2\right)$$

$$+ \frac{8(n-1)(n-2)}{p}\text{Cov}\left(x_{i1}x_{j1}x_{i2}x_{j2}, x_{k1}^2 x_{l3}^2\right).$$

Again, these are the same nine covariance expressions as for the variances.

### 3.4  Calculation of the appearing covariance expressions

In the variance and covariance formulas above, only nine different covariance expressions appear. Assuming normality, calculations in the eigenbasis simplify considerably: under normality, uncorrelatedness implies independence.

### 1. covariance expression

$$\sum_{ijkl}\text{Cov}\big(y_{i1}^2 y_{j1}^2, y_{k1}^2 y_{l1}^2\big) = \sum_{ijkl}\mathbb{E}[y_{i1}^2 y_{j1}^2 y_{k1}^2 y_{l1}^2] - \mathbb{E}[y_{i1}^2 y_{j1}^2]\mathbb{E}[y_{k1}^2 y_{l1}^2]$$

$$= \begin{cases} \sum_i \mathbb{E}[y_i^8] - \mathbb{E}^2[y_i^4] = \sum_i 105\gamma_i^4 - 9\gamma_i^4 = 96\sum_i \gamma_i^4 & i = j = k = l \text{ (case 1)} \\ 4\sum_{i,j\neq i}\mathbb{E}[y_i^2]\mathbb{E}[y_j^6] - \mathbb{E}[y_i^2]\mathbb{E}[y_j^2]\mathbb{E}[y_j^4] = 4\sum_{i,j\neq i}\gamma_i 15\gamma_j^3 - \gamma_i 3\gamma_j^3 & i \neq j = k = l \text{ (case 2)} \\ \qquad = 48\sum_{i,j\neq i}\gamma_i\gamma_j^3 & \\ 0 & i = j \neq k = l \text{ (case 3)} \\ 2\sum_{ij\neq i}\mathbb{E}[y_i^4]\mathbb{E}[y_j^4] - \mathbb{E}^2[y_i^2]\mathbb{E}^2[y_j^2] = 2\sum_{ij\neq i} 9\gamma_i^2\gamma_j^2 - \gamma_i^2\gamma_j^2 & i = k \neq j = l \text{ (case 4)} \\ \qquad = 16\sum_{ij\neq i}\gamma_i^2\gamma_j^2 & \\ 4\sum_{i,j\neq i,k\neq i,j}\mathbb{E}[y_i^2]\mathbb{E}[y_k^2]\mathbb{E}[y_j^4] - \mathbb{E}[y_i^2]\mathbb{E}[y_k^2]\mathbb{E}^2[y_j^2] & i,k,j \text{ distinct}, j = l \text{ (case 5)} \\ \qquad = 4\sum_{i,j\neq i,k\neq i,j} 3\gamma_i\gamma_j^2\gamma_k - \gamma_i\gamma_j^2\gamma_k = 8\sum_{i,j\neq i,k\neq i,j}\gamma_i\gamma_j^2\gamma_k & \\ 0 & i,j,k,l \text{ distinct (case 6)} \end{cases}$$

**2. covariance expression**

$$\sum_{ijkl} \mathrm{Cov}\big(y_{i1}^2 y_{j1}^2, y_{k1}y_{l1}y_{k2}y_{l2}\big) = \mathbb{E}[y_{i1}^2 y_{j1}^2 y_{k1}y_{l1}]\mathbb{E}[y_{k2}y_{l2}] - \mathbb{E}[y_{i1}^2 y_{j1}^2]\mathbb{E}[y_{k1}y_{l1}]\mathbb{E}[y_{k2}y_{l2}]$$

The expression is zero when $k = l$ does not hold. Four cases remain:

$$= \begin{cases} \sum_i \big(\mathbb{E}[y_i^6] - \mathbb{E}[y_i^4]\mathbb{E}[y_i^2]\big)\mathbb{E}[y_i^2] = \sum_i 15\gamma_i^4 - 3\gamma_i^4 = 12\sum_i \gamma_i^4 & i = j = k = l \text{ (case 1)} \\ 2\sum_{i,j\neq i}\mathbb{E}[y_i^2]\mathbb{E}[y_j^4]\mathbb{E}[y_j^2] - \mathbb{E}[y_i^2]\mathbb{E}^3[y_j^2] = 2\sum_{i,j\neq i}\gamma_i 3\gamma_j^3 - \gamma_i\gamma_j^3 & i \neq j = k = l \text{ (case 2)} \\ \qquad\qquad = 4\sum_{i,j\neq i}\gamma_i\gamma_j^3 & \\ 0 & i, k, j \text{ distinct}, k = l \text{ (case 3)} \\ 0 & i \neq j \neq k = l \text{ (case 4)} \end{cases}$$

**3. covariance expression**

$$\sum_{ijkl} \mathrm{Cov}\big(y_{i1}y_{j1}y_{i2}y_{j2}, y_{k1}y_{l1}y_{k2}y_{l2}\big) = \mathbb{E}[y_{i1}y_{j1}y_{k1}y_{l1}]\mathbb{E}[y_{i2}y_{j2}y_{k2}y_{l2}] - \mathbb{E}[y_{i1}y_{j1}]\mathbb{E}[y_{i2}y_{j2}]\mathbb{E}[y_{k1}y_{l1}]\mathbb{E}[y_{k2}y_{l2}]$$

The expression is zero if the indices are not paired. Three cases exist:

$$= \begin{cases} \sum_i \mathbb{E}^2[y_i^4] - \mathbb{E}^4[y_i^2] = \sum_i 9\gamma_i^4 - \gamma_i^4 = 8\sum_i \gamma_i^4 & i = j = k = l \text{ (case 1)} \\ 0 & i = j \neq k = l \text{ (case 2)} \\ 2\sum_{i,j\neq i}\mathbb{E}^2[y_i^2]\mathbb{E}^2[y_j^2] = 2\sum_{i,j\neq i}\gamma_i^2\gamma_j^2 & i = k \neq j = l \text{ (case 3)} \end{cases}$$

**4. covariance expression**

$$\sum_{ijkl} \mathrm{Cov}\big(y_{i1}y_{j1}y_{i2}y_{j2}, y_{k1}y_{l1}y_{k3}y_{l3}\big) = \mathbb{E}[y_{i1}y_{j1}y_{k1}y_{l1}]\mathbb{E}[y_{i2}y_{j2}]\mathbb{E}[y_{k3}y_{l3}] - \mathbb{E}[y_{i1}y_{j1}]\mathbb{E}[y_{i2}y_{j2}]\mathbb{E}[y_{k1}y_{l1}]\mathbb{E}[y_{k3}y_{l3}]$$

$$= \Big(\mathbb{E}[y_{i1}y_{j1}y_{k1}y_{l1}] - \mathbb{E}[y_{i1}y_{j1}]\mathbb{E}[y_{k1}y_{l1}]\Big)\mathbb{E}[y_{i2}y_{j2}]\mathbb{E}[y_{k3}y_{l3}]$$

The expression is zero, if either $i = j$ or $k = l$ do not hold, or the first expection separates. Therefore, only the case $i = j = k = l$ remains:

$$= \sum_i \big(\mathbb{E}[y_i^4] - \mathbb{E}^2[y_i^2]\big)\mathbb{E}^2[y_i^2] = \sum_i \big(3\gamma_i^2 - \gamma_i^2\big)\gamma_i^2 = 2\sum_i \gamma_i^4.$$

**5. covariance expression**

$$\sum_{ijkl} \mathrm{Cov}\big(x_{i1}^2 x_{j1}^2, x_{k1}^2 x_{l2}^2\big) = \mathbb{E}[x_{i1}^2 x_{j1}^2 x_{k1}^2 x_{l2}^2] - \mathbb{E}[x_{i1}^2 x_{j1}^2]\mathbb{E}[x_{k1}^2 x_{l2}^2]$$

$$= \Big(\mathbb{E}[x_{i1}^2 x_{j1}^2 x_{k1}^2] - \mathbb{E}[x_{i1}^2 x_{j1}^2]\mathbb{E}[x_{k1}^2]\Big)\mathbb{E}[x_{l2}^2]$$

$$= \begin{cases} \sum_{ij}\big(\mathbb{E}[y_i^6] - \mathbb{E}[y_i^4]\mathbb{E}[y_i^2]\big)\mathbb{E}[y_l^2] = \sum_{il}(15\gamma_i^3 - 3\gamma_i^3)\gamma_l = 12\sum_{ij}\gamma_i^3\gamma_l & i = j = k \text{ (case 1)} \\ 2\cdot\sum_{i,j\neq i,l}\big(\mathbb{E}[y_i^4]\mathbb{E}[y_j^2] - \mathbb{E}^2[y_i^2]\mathbb{E}[y_j^2]\big)\mathbb{E}[y_l^2] = 4\sum_{i,j\neq i,l}\gamma_i^2\gamma_j\gamma_l & i = k \neq j \text{ (case 2)} \end{cases}$$

**6. covariance expression**

$$\sum_{ijkl} \mathrm{Cov}\big(x_{i1}^2 x_{j2}^2, x_{k1}^2 x_{l2}^2\big) = \mathbb{E}[x_{i1}^2 x_{j2}^2 x_{k1}^2 x_{l2}^2] - \mathbb{E}[x_{i1}^2 x_{j2}^2]\mathbb{E}[x_{k1}^2 x_{l2}^2]$$

$$= \mathbb{E}[x_{i1}^2 x_{k1}^2]\mathbb{E}[x_{j2}^2 x_{l2}^2] - \mathbb{E}[x_{i1}^2]\mathbb{E}[x_{j2}^2]\mathbb{E}[x_{k1}^2]\mathbb{E}[x_{l2}^2]$$

$$= \begin{cases} \mathbb{E}[x_{i1}^4]\mathbb{E}[x_{j2}^4] - \mathbb{E}^2[x_{i1}^2]\mathbb{E}^2[x_{j2}^2] = 8\sum_{ij}\gamma_i^2\gamma_j^2 & i = k, j = l \text{ (case 1)} \\ 2\cdot 3\sum_{i,j\neq l}\gamma_i^2\gamma_j\gamma_l & i = k, j \neq l \text{ (case 2)} \end{cases}$$

**7. covariance expression**

$$\sum_{ijkl} \text{Cov}\left(x_{i1}^2 x_{j2}^2, x_{k1}^2 x_{l3}^2\right) = \mathbb{E}[x_{i1}^2 x_{j2}^2 x_{k1}^2 x_{l3}^2] - \mathbb{E}[x_{i1}^2 x_{j2}^2]\mathbb{E}[x_{k1}^2 x_{l3}^2]$$

$$= \left(\mathbb{E}[x_{i1}^2 x_{k1}^2] - \mathbb{E}[x_{i1}^2]\mathbb{E}[x_{k1}^2]\right)\mathbb{E}[x_{j2}^2]\mathbb{E}[x_{l3}^2]$$

$$= 3\sum_{ijl} \gamma_i^2 \gamma_j \gamma_l$$

**8. covariance expression**

$$\sum_{ijkl} \text{Cov}\left(x_{i1}x_{j1}x_{i2}x_{j2}, x_{k1}^2 x_{l2}^2\right) = \mathbb{E}[x_{i1}x_{j1}x_{i2}x_{j2}, x_{k1}^2 x_{l2}^2] - \mathbb{E}[x_{i1}x_{j1}x_{i2}x_{j2}]\mathbb{E}[x_{k1}^2 x_{l2}^2]$$

$$= \mathbb{E}[x_{i1}x_{j1}x_{k1}^2]\mathbb{E}[x_{i2}x_{j2}x_{l2}^2] - \mathbb{E}^2[x_{i1}x_{j1}]\mathbb{E}[x_{k1}^2]\mathbb{E}[x_{l2}^2]$$

The expression is only non-zero when $i = j$ holds. In addition, either $k = i = j$ or $l = i = j$ has to hold.

$$= \begin{cases} \sum_i \mathbb{E}^2[x_i^4] - \mathbb{E}^4[x_i^2] = 8\sum_i \gamma_i^4 & i = j = k = l \text{ (case 1)} \\ \sum_{i,l\neq i} \mathbb{E}[x_i^4]\mathbb{E}[x_i^2]\mathbb{E}[x_l^2] - \mathbb{E}^3[x_i^2]\mathbb{E}[x_l^2] = 6\sum_{i,l\neq i} \gamma_i^3 \gamma_l & i = j = k \neq l \text{ (case 2)} \end{cases}$$

**9. covariance expression**

$$\sum_{ijkl} \text{Cov}\left(x_{i1}x_{j1}x_{i2}x_{j2}, x_{k1}^2 x_{l3}^2\right) = \mathbb{E}[x_{i1}x_{j1}x_{i2}x_{j2}x_{k1}^2 x_{l3}^2] - \mathbb{E}[x_{i1}x_{j1}x_{i2}x_{j2}]\mathbb{E}[x_{k1}^2 x_{l3}^2]$$

$$= \left(\mathbb{E}[x_{i1}x_{j1}x_{k1}^2] - \mathbb{E}[x_{i1}x_{j1}]\mathbb{E}[x_{k1}^2]\right)\mathbb{E}[x_{i2}x_{j2}]\mathbb{E}[x_{l3}^2]$$

The expression is only non-zero if $i = j = k$ holds:

$$= \sum_{il} \left(\mathbb{E}[y_i^4] - \mathbb{E}^2[y_i^2]\right)\mathbb{E}[y_i^2]\mathbb{E}[y_l^2]$$

$$= \sum_{ij} \left(3\gamma_i^2 - \gamma_i^2\right)\gamma_i\gamma_j = 2\sum_{ij} \gamma_i^3 \gamma_j.$$