[Reviews · NeurIPS 2013]

Submitted by Assigned_Reviewer_5

Summary: Analytic shrinkage means using a known formula for the optimal strength of regularization for an estimated covariance matrix. This paper shows that the known formula is valid under weaker conditions than in the original proof. An algorithm for better practical shrinkage is provided also.

Quality: High.

Clarity: Good.

Originality: Good, assuming that the central result really is new.

Significance: Mathematically significant. Of limited practical importance, because the recommended regularization strength remains the same, and experiments show limited benefit for the new AOC method.


DETAILED COMMENTS

19: "implies" should be "requires" as on line 45.

90: Intuitively, from where do the very high order moments arise? 8 here and 16 later. Should we take these seriously in practice?

291: This iterated process is the main practical contribution. Please provide more detail, maybe pseudocode. Figure 3 can be smaller to make space.

302: "percentual" is not a word and in any case should be "fraction."

373: What factor model is used here? Explain. Is it significant that AOC does better than the factor model?

386 to 388: Discuss why QDA accuracy is so variable.

396: Discuss why LDA (shrinkage) is worse than plain LDA.
Summary: Good theoretical paper.

Submitted by Assigned_Reviewer_6

The authors focus on the problem of the estimation of covariance matrices using empirical estimators. To avoid overfitting in high-dimensional cases they shrink the covariance matrix towards a multiple of the identity matrix. The optimal amount of shrinkage \lambda can be estimated empirically by minimizing the expected error of the shrinkage estimate. Then they show that existing results about the consistency of the estimate of \lambda depend on an assumption which does not hold in practice for most real-world datasets. The authors then introduce a new wider set of assumptions and proof that they guarantee consistency. Also, they show that in certain cases there will be no asymptotic shrinkage of the empirical estimate of the covariance matrix. As a solution that they propose is then to perform the shrinkage in the orthogonal complement to the eigen vector with largest eigen value. Since a dataset may have multiple large eigenvalues they iterate the procedure and automatically select the number of retained eigen directions. A series of experiments with synthetic and real-world data validate the proposed approach.

Quality:

The paper seems to be sound, with the experiments validating the theoretical results. I did not verified the proofs of the main theorems since these are in the supplementary material, which I am not required to examine. The experimental evaluation seems to be thorough.

Clarity:

The paper is clearly written and well organized. However, the main document does not describe how the orthogonal complement shrinkage is actually performed and the motivation for this approach is also not strong enough in the paper. The authors should clarify how this technique is implemented in practice.

Originality:

I find interesting and worth on its own the contribution of the authors to show that the existing shrinkage techniques will not produce enough shrinkage in some cases. The proposed orthogonal complement shrinkage is novel up to my knowledge. However, it is a bit incremental since it represents only a modification of already existing approaches.

Significance:

In my opinion, the most significant contribution is to point out the failure of the shrinkage method in several relevant cases, as well as the theoretical contributions contained in the paper. The orthogonal complement shrinkage seems to be less significant than these other elements.
Summary: A technically sound and well written paper that presents some interesting theoretical results and proposes an incremental contribution to improve existing techniques.

Submitted by Assigned_Reviewer_7

This paper makes several contributions in the area of analytic shrinkage for regularized covariance matrix estimation.

First the authors show that an assumption on the covariance structure (bound on the 8th moments in the eigenbasis) that is used to prove the consistency of analytic shrinkage does not hold in several real datasets from different fields by deriving two checkable consequences of this result:
(i) the largest eigenvalue grows with the fourth root
(ii) the eigenvalue dispersion is bounded

The authors then provide weaker assumptions that do not place any constraints on the covariance structure and show that analytic shrinkage is still consistent under these weaker assumptions. They also provide an expected result showing a problem with the limit behavior of analytic shrinkage that probably applies to many real world datasets.

Next the authors provide an alternative shrinkage based procedure for regularized covariance matrix estimation they call oc-shrinkage. In this method shrinkage is only applied to the orthogonal complement of the first (or first few) eigenvectors. They show this procedure still results in a consistent estimator and empirically compare it to analytic shrinkage in several different domains. Oc-shrinkage outperforms analytic shrinkage in each case.

This paper contains novel and theoretically sensible ideas for regularized covariance matrix estimation which have very clear important real world applications. The results appear to be correct and rigorous (I was not able to check all of the proofs) and validated by the empirical results.

The paper is well written and clear for the most part. I think section 5 could be improved to provide more details. While the basic idea of oc-shrinkage is clear, I'm not sure how all the details in the procedure work out in practice. Since this is probably the most important contribution (at least most important deliverable) of the paper, a step by step procedure for implementing oc-shrinkage in practice would be useful.
Summary: The authors provide new results which generalize analytic shrinkage for regularized covariance matrix estimation to arbitrary covariance structures and provide a novel alternative procedure that appears to outperform the standard analytic shrinkage procedure while retaining consistency. The ideas presented are sensible and rigorously justified and the novel alternative procedure could lead to significant improvements in covariance estimation in many real world applications.
Author Feedback

Author rebuttal: First of all we would like to thank the reviewers for their helpful and encouraging comments to our manuscript.

We see as a recurring comment raised by the reviewers, that our novel shrinkage proceedure should be described in a bit more detail so that it can be more easily implemented in practice. We think that
this is a really helpful point: in the revision we will include a more detailed description in the
main text and add pseudo-code.

For the information of the reviewers, we provide the pseudocode below.

The ingredients of aoc-shrinkage are
- calc_lambda(X):
returns the standard shrinkage strength
- calc_Delta_R(X,lambda_1,nu_1,lambda_2,nu_2):
returns an estimate of EMSE (cmp. eq. 3) improvement of lambda_1, nu_1 over lambda_2,nu_2
(( C_shr = (1-lambda)*S + lambda*nu*eye(p) ))
- calc_sigma(X,lambda_1,nu_1,lambda_2,nu_2):
returns an estimate of the std. dev. of Delta_R
We agree that the formulas for Delta_R and sigma have to be put explicitly in the text/supplemental.

Let S be the covariance matrix, V the matrix of eigenvectors sorted with
descending eigenvalues and l the vector of eigenvalues:
1. r* = 0, r = 1
2. lambda_old = calc_lambda(X)
3. nu_old = mean(l)
4. while true
______# shrinkage on the orthogonal complement
______# of r directions:
5. ____X_oc = V(:,r+1:p)'*X
6. ____lambda_new = calc_lambda( X_oc )
7. ____nu_new = mean( l(r+1:p) )
______# calculate estimate and std. dev. of
______#exp. mean squared error difference:
8. ____Delta_R = calc_Delta_R(X_oc, ________lambda_new,nu_new,lambda_old,nu_old)
9. ____sigma_R = calc_sigma(X_oc, ________lambda_new,nu_new,lambda_old,nu_old)
______# if there is an improvement,
______#accept r and continue. else stop.
10. ___if Delta_R - m * sigma_R > 0
11. ________r* = r* + 1
12. ________r = r + 1
13. ________lambda_old = lambda_new
14. ________nu_old = nu_new
15. ___else
16. ________break
17. ___end
18. end
19. P_pca = V(:,1:r*)*(V(:,1:r*)'
20. P_oc = V(:,r*+1:p)*V(:,r*+1:p)'
21. C_shr = (1-lambda_old)*S + lambda_old*eye(p)*nu_old
22. return P_pca*S*P_pca' + P_oc*C_shr*P_oc'